# Reconstruction on Trees and Low-Degree Polynomials

**Frederic Koehler**[*]
Stanford University
fkoehler@stanford.edu

**Elchanan Mossel**[†]
Massachusetts Institute of Technology
elmos@mit.edu

## Abstract

The study of Markov processes and broadcasting on trees has deep connections to a variety of areas including statistical physics, graphical models, phylogenetic reconstruction, Markov Chain Monte Carlo, and community detection in random graphs. Notably, the celebrated Belief Propagation (BP) algorithm achieves Bayes-optimal performance for the reconstruction problem of predicting the value of the Markov process at the root of the tree from its values at the leaves.

Recently, the analysis of low-degree polynomials has emerged as a valuable tool for predicting computational-to-statistical gaps. In this work, we investigate the performance of low-degree polynomials for the reconstruction problem on trees. Perhaps surprisingly, we show that there are simple tree models with $N$ leaves and bounded arity where (1) nontrivial reconstruction of the root value is possible with a simple polynomial time algorithm and with robustness to noise, but not with any polynomial of degree $N^c$ for $c > 0$ a constant depending only on the arity, and (2) when the tree is unknown and given multiple samples with correlated root assignments, nontrivial reconstruction of the root value is possible with a simple Statistical Query algorithm but not with any polynomial of degree $N^c$. These results clarify some of the limitations of low-degree polynomials vs. polynomial time algorithms for Bayesian estimation problems. They also complement recent work of Moitra, Mossel, and Sandon who studied the circuit complexity of Belief Propagation. As a consequence of our main result, we are able to prove a result of independent interest regarding the performance of RBF kernel ridge regression for learning to predict the root coloration: for some $c' > 0$ depending only on the arity, $\exp(N^{c'})$ many samples are needed for the kernel regression to obtain nontrivial correlation with the true regression function (BP). We pose related open questions about low-degree polynomials and the Kesten-Stigum threshold.

## 1 Introduction

Understanding the computational complexity of random instances has been the goal of an extensive line of research spanning multiple decades and different research areas such as cryptography, high-dimensional statistics, complexity theory, and statistical physics. In particular, this includes work on satisfiability and refutation of random constraint satisfaction problems and on computational-to-statistical gaps. In much of this work, evidence for computational hardness is indirect because there are well-known barriers to proving hardness from classical worst-case assumptions such as

---

[*]Supported in part by E. Mossel's Vannevar Bush Faculty Fellowship ONR-N00014-20-1-2826, NSF award CCF-1704417, NSF Award IIS-1908774, and N. Anari's Sloan Research Fellowship. Part of this work was completed while at MIT and while participating in the program *Computational Complexity of Statistical Inference* at the Simons Institute for the Theory of Computing.

[†]Supported by Simons-NSF collaboration on deep learning NSF DMS-2031883, by Vannevar Bush Faculty Fellowship award ONR-N00014-20-1-2826 and by a Simons Investigator Award in Mathematics (622132)

NP-hardness (Feigenbaum and Fortnow 1993; Akavia et al. 2006; Bogdanov and Trevisan 2006; Applebaum et al. 2008).

Recently, low-degree polynomials have emerged as a powerful tool for predicting computational-to-statistical gaps. Computational-to-statistical gaps are situations where it is impossible for polynomial time algorithms to estimate a desired quantity of interest from the data, even though computationally inefficient ("information-theoretic") algorithms can succeed at the same task. Heuristics based on low-degree polynomials have especially been used in the context of Bayesian estimation and testing problems and partially motivated by connections with (lower bounds for) the powerful Sum-of-Squares proof system. More specifically, a recent line of work (e.g. Hopkins and Steurer 2017; Hopkins 2018; Kunisky et al. 2019; Bandeira et al. 2020; Gamarnik et al. 2020; Holmgren and Wein 2020; Wein 2020; Bresler and Huang 2021; Mao and Wein 2021) showed that a suitable "low-degree heuristic" can be used to predict computational-statistical gaps for a variety of problems such as recovery in the multicommunity stochastic block model, sparse PCA, tensor PCA, the planted clique problem, certification in the zero-temperature Sherrington-Kirkpatrick model, the planted sparse vector problem, and for finding solutions in random $k$-SAT problems. Furthermore, it was observed that the predictions from this method generally agree with those conjectured using other techniques (for example, statistical physics heuristics based on studying BP/AMP fixed points, see e.g. Decelle et al. 2011; Deshpande and Montanari 2015; Mohanty et al. 2021). Some of the merits of the low-degree polynomial framework include that it is relatively easy to use (e.g. compared to proving SOS lower bounds), and that low degree polynomials capture the power of the "local algorithms" framework used in e.g. (Gamarnik and Sudan 2014; Chen et al. 2019) as well as algorithms which incorporate global information, such as spectral methods or a constant number of iterations of Approximate Message Passing (Wein 2020).

In this work, we investigate the power of low-degree polynomials for the (average case) reconstruction problem on trees. We define the model and results in the next sections, but first give an informal summary. The goal for reconstruction on trees is to estimate the value of the Markov process at the root given its value at the leaves (in the limit where the depth of the tree goes to infinity), and two key parameters of the model are the arity of the tree $d$ and the magnitude of the second eigenvalue $\lambda_2$ of the broadcast chain. Importantly, when $d|\lambda_2|^2 > 1$ it is known (Kesten and Stigum 1966) that nontrivial reconstruction of the root is possible just from knowing the counts of the leaves of different types, whereas when $d|\lambda_2|^2 < 1$ such count statistics have no mutual information with the root (but more complex statistics of the leaves may) Mossel and Peres 2003. This threshold $d|\lambda_2|^2 = 1$ is known as the *Kesten-Stigum threshold* (Kesten and Stigum 1966) and it plays a fundamental role in other problems, such as algorithmic recovery in the stochastic block model (Abbe 2017) and phylogenetic reconstruction (Daskalakis et al. 2006). Count statistics can be viewed as degree 1 polynomials of the leaves, which begs the question of what information more general polynomials can extract from the leaves.

In this paper, we answer this question in the limit case $\lambda_2 = 0$. Perhaps surprisingly, we find that the Kesten-Stigum threshold remains tight in the sense that even polynomials of degree $N^c$ for a small $c > 0$ are not able to correlate with the root label (Theorem 6), whereas computationally efficient reconstruction is generally possible as long as $d$ is a sufficiently large constant (Theorem 5) and even when a constant fraction of leaves are replaced by noise. Building on the polynomial lower bound, we prove superpolynomiallly (in fact, subexponentially) many samples are needed for Gaussian Kernel Ridge Regression (KRR) to (weakly) learn to regression function which predicts the root from the leaf colorations (Theorem 20). This gives a simple and natural model where KRR provably fails that is outside the reach of existing lower bounds such as (Kamath et al. 2020).

We also consider an analogous question where the tree is unknown, and the algorithm has access to $m$ i.i.d. samples of the Markov process where the root is biased towards an unknown label $Y^*$. In this setting, polynomials of degree $N^c$ again fail to correlate with the label $Y^*$, but we show that a simple algorithm, straightforwardly implementable in the Statistical Query (SQ) model (Kearns 1998), can recover $Y^*$ in polynomial time (Theorem 11). Together, these results show that low-degree polynomials behave very differently in our setting than one might intuit based on previous work in related settings, such as in random constraint satisfaction problems or the block model.

## 1.1 Preliminaries

**Notation.** We use the standard notation $O_a(\cdot)$ to denote an upper bound with an implied constant which is allowed to depend on $a$; the notation $\text{poly}_a(\cdot)$ is similar for denoting a bound which is polynomial in its parameters. We let $d_{TV}(P, Q)$ denote the total variation distance between distributions $P$ and $Q$ normalized to be in $[0, 1]$ and we use $I(X; Y \mid Z)$ for the conditional mutual information of random variables $X, Y$ conditional on $Z$; see (Cover 1999). Given a vector $x$ and a subset of coordinates $S$, we let $x_S$ denote the $|S|$-dimensional vector corresponding to elements of $S$.

**Markov processes on trees.** We consider Markov processes with state space $\Sigma = [q] = \{1, \ldots, q\}$ where $q \geq 1$ is the size of the alphabet. Let $M : q \times q$ be the transition matrix of a time-homogeneous Markov chain on $\Sigma$, also referred to as the *broadcast channel*. For simplicity, we always assume henceforth that $M$ is *ergodic* (irreducible and aperiodic, see (Durrett 2019)) so it has a unique stationary distribution $\pi_M$. Let $T = (V, E, \rho)$ be a rooted tree with vertex set $V$, root $\rho \in V$, and where $E$ is the set of directed edges $(u, v)$ where $u$ is the parent of $v$ in the corresponding tree. The *broadcast process* on tree $T$ of depth $\ell$ with transition matrix $M$ and root prior $\nu$ a probability measure on $[q]$ is given by

$$\mu_{\ell,\nu}(x) := \nu(x_\rho) \prod_{(u,v) \in E} M_{x_u, x_v}.$$

When not otherwise indicated, $\nu$ is the stationary distribution for $M$. The probability measure $\mu$ is a *Markov Random Field* on the tree $T$. This means that if $A, B$ are subsets of the vertices of $T$ and all paths in $T$ from $A$ to $B$ pass through a third set of vertices $S$, then $X_A$ and $X_B$ are conditionally independent given $X_S$. This is called the *Markov property*, see e.g. (Lauritzen 1996). In this paper, we focus on the setting of complete $d$-ary trees (i.e. trees where every non-leaf node has $d$ children, and all leaf nodes are at the same depth). For the $d$-ary tree of depth $\ell \geq 0$, we let $L$ be the set of leaves of the tree, i.e. the set of vertices in the tree at depth $\ell$.

**Definition 1.** We say that *reconstruction is possible* on the $d$-ary tree with channel $M$ if

$$\inf_{\ell \geq 1} \max_{c, c' \in [q]} d_{TV}\Big(\mathcal{L}_{\mu_\ell}(X_L \mid X_\rho = c), \mathcal{L}_{\mu_\ell}(X_L \mid X_\rho = c')\Big) > 0$$

where the notation $\mathcal{L}_\mu(X|E)$ denotes the conditional law of $X$ under $\mu$ given event $E$ occurs, $\mu_\ell$ is the corresponding broadcast process on the depth $\ell$ tree with root $\rho$ and $L = L_\ell$ is the set of leaves.

When reconstruction is possible, the Bayes-optimal estimate of the root given the leaves can be computed in linear time by passing messages up the tree using the Belief Propagation algorithm (Mezard and Montanari 2009); for our purposes, we will not need the explicit formula for BP, which can be derived by applying Bayes rule, but refer the interested reader to the reference.

Given a matrix $M$, we let $\lambda_2(M)$ denote the second-largest eigenvalue of $M$ in absolute value. The *Kesten-Stigum (KS) threshold* on the $d$-ary tree is given by the equation $d|\lambda_2(M)|^2 = 1$. Building upon the original work of (Kesten and Stigum 1966), it was shown that the KS threshold is sharp for the problem of *count reconstruction* on trees (Mossel and Peres 2003): count reconstruction is possible when $d|\lambda_2(M)|^2 > 1$ and impossible when $d|\lambda_2(M)|^2 < 1$.

**Definition 2.** Let $C(x) := (\#\{i : x_i = c\})_{c \in [q]}$ be the function which computes count statistics of an input vector $x$ with entries in $[q]$. We say that *count-reconstruction is possible* on the $d$-ary tree with channel $M$ if $\inf_{\ell \geq 1} \max_{c, c' \in [q]} d_{TV}(\mathcal{L}_{\mu_\ell}(C(X_L) \mid X_\rho = c), \mathcal{L}_{\mu_\ell}(C(X_L) \mid X_\rho = c')) > 0$ where the notation $\mathcal{L}(X|E)$ denotes the conditional law of $X$ given event $E$, $\mu_\ell$ is the corresponding broadcast process on the depth $\ell$ tree and $L = L_\ell$ is the set of leaves on this tree.

Next, we define a notion of noisy reconstruction which plays an important role in this paper:

**Definition 3.** For $\epsilon \in (0, 1)$, we say that *$\epsilon$-noisy reconstruction is possible* on the $d$-ary tree with channel $M$ if $\inf_{\ell \geq 1} \max_{c, c'} d_{TV}(\mathcal{L}_{\mu_\ell}(X'_L = \cdot \mid X_\rho = c), \mathcal{L}_{\mu_\ell}(X'_L = \cdot \mid X_\rho = c')) > 0$ where $X'$ is the $\epsilon$-noisy version of the broadcast process values $X$, generated by independently for each vertex $v$, setting $(X'_L)_v = (X_L)_v$ with probability $1 - \epsilon$ and otherwise sampling $(X'_L)_v$ from $Uni([q])$[3].

---

[3]More generally, our results hold where the noise is from any full support distribution on $[q]$.

Note that in this definition, the law of $X'_L \mid X_\rho = c$ can equivalently be written as $\mathcal{L}_{\mu_\ell}(X_L \mid X_\rho = c)T_\epsilon$ where $T_\epsilon$ is the usual noise operator that independently resamples each coordinate of its input vector with probability $\epsilon$, see e.g. O'Donnell [2014]; Hopkins [2018]. Finally, we recall from (Janson and Mossel [2004]) the following standard definition: we say that *robust reconstruction* on the $d$-ary tree with channel $M$ is possible if $\epsilon$-noisy reconstruction is possible *for every* $\epsilon \in (0,1)$.

**Low-degree polynomials and computational-statistical gaps.** As discussed in the introduction, low degree polynomials have been studied in a wide variety of contexts and settings. The recent work (Schramm and Wein [2020]) showed that a version of the low-degree polynomial heuristic can predict the recovery threshold for natural Bayesian estimation problems, even when the recovery threshold is below the detection/testing threshold. In the present work, we will use the following key definition from their paper[4]

**Definition 4** (Degree-$D$ Maximum Correlation (Schramm and Wein [2020])). Suppose that $(X, Y) \sim P$ where $X$ is a random vector in $\mathbb{R}^N$ and $Y$ is a random variable valued in $\mathbb{R}$. The *degree-$D$ maximum correlation* is defined to be

$$\mathrm{Corr}_{\leq D}(P) := \sup_{f \in \mathbb{R}[X]_{\leq D}, \mathbb{E}_P[f(X)^2] \neq 0} \frac{\mathbb{E}_P[f(X) \cdot Y]}{\sqrt{\mathbb{E}_P[f(X)^2]}}$$

where $\mathbb{R}[X]_{\leq D}$ is the space of degree at most $D$ multivariate polynomials in variables $X_1, \ldots, X_N$ with real-valued coefficients.

As explained there, when the target label $Y^*$ is a vector this definition can be applied with $Y$ equal to each of the coordinates of $Y^*$. We note that we could rephrase our results in terms of a testing problem (as in much of the prior work on the low-degree method), but the above definition is more natural in our context (it avoids the need to introduce a "null distribution" $Q$). In what follows, we omit the distribution $P$ the expectation is taken over as long as it is clear from context.

Instead of referring to polynomial degree directly, we usually use the following more convenient and equivalent definition. Suppose $f$ is a function $[q]^n \to \mathbb{R}$. We define the (Efron-Stein) *degree* of $f$ to be the minimal $D$ such that there exist functions $f_S : [q]^{|S|} \to \mathbb{R}$ so that $f(x) = \sum_{S \subset [n], |S| \leq D} f_S(x_S)$. One such minimal choice of $f_S$ is the Efron-Stein decomposition over $Uni[q]^n$, see e.g. (O'Donnell [2014]); this notion is also equivalent to the minimal degree polynomial representing $f$ where the variables are the one-hot encoding $x \mapsto (\mathbb{1}(x_i = c))_{i \in [n], c \in [q]}$.

**Reconstruction below the KS threshold.** In this paper, we will largely consider the problem of tree reconstruction with matrices $M$ with $\lambda_2(M) = 0$; these exactly correspond to Markov chains which mix perfectly within a bounded number of steps. (There are many examples of such chains, for concreteness we give a very small example below in Example [13]. Obviously, for such a chain $M$, $d|\lambda_2|^2 = 0$ for any value of $d$ so such a model is always below the Kesten-Stigum threshold. Nevertheless, based on general results from existing work we know that near-perfect reconstruction of the root is possible (e.g. using Belief Propagation, which computes the exact posterior distribution (Mezard and Montanari [2009])). This is true as long as $d$ is sufficiently large, and even with a constant amount of noise $\epsilon$:

**Theorem 5** (Mossel and Peres [2003], Theorem [22] below). *Suppose $M$ is a the transition matrix of a Markov chain with pairwise distinct rows[5] i.e. for all $i, j \in [q]$ the rows $M_i$ and $M_j$ are distinct vectors. Let $\delta \in (0,1)$ be arbitrary. There exists $d_0 = d_0(M, \delta)$ and $\epsilon > 0$ such that for all $d \geq d_0$, $\epsilon$-noisy reconstruction is possible on the $d$-ary tree and furthermore there exists a polynomial-time computable function $f = f_{M,\ell}$ valued in $[q]$ such that*

$$\max_{c \in [q]} \mathrm{Pr}(f(X'_L) \neq X_\rho \mid X_\rho = c) < \delta$$

*where $X'_L$ is the $\epsilon$-noisy version of $X_L$ (see Definition [3]).*

This exact statement does not appear in Mossel and Peres [2003] but follows from arguments presented there; for completeness, we include a proof in the appendix (Theorem [22]). From the proof, we can see that a very simple recursive estimator is enough to solve this problem.

---

[4]In our notation $X$ and $y$ are swapped compared to theirs, to match the convention in the broadcast process.

[5]This condition is needed to rule out the case of e.g. a rank one matrix $M$ where reconstruction is clearly impossible. See also (Mossel and Peres [2003]) for a more complex and precise condition.

## 1.2 Our Results

We study the power of low-degree polynomials for the problem of reconstructing the root of a Markov process. We consider this question in the context of two very closely related versions of the model which have both been extensively studied in the literature.

*Reconstruction with a known tree.* In this setting, the algorithm is given access to the leaf values from a single realization of the Markov process, and the goal is to estimate the root (where we are going to be interested in estimators which are low-degree polynomials of the leaves). The tree structure is known and the estimator/polynomial is allowed to depend on this information directly.

*Reconstruction with an unknown tree.* In this setting, the data is still generated by a complete $d$-ary tree but the tree (in other words, the true ordering of the leaves) is unknown to the algorithm. This version of the model has been extensively studied due to close connections to the problem of phylogenetic reconstruction in biology, see e.g. Felsenstein 2004; Daskalakis et al. 2006; Steel 2016. Because this task is more difficult information-theoretically[6], the algorithm is given access to $m$ i.i.d. samples from the broadcast model; we give a more precise definition of the model below.

**Results for reconstruction with a known tree.** We consider the problem of tree reconstruction with matrices $M$ with $\lambda_2(M) = 0$; these exactly correspond to Markov chains which mix perfectly within a bounded number of steps. As discussed above in Preliminaries, while such models are always below the Kesten-Stigum threshold for any value of the the arity $d$, under fairly weak conditions on $M$ the reconstruction problem is still solvable for $d$ sufficiently large (Theorem 5). This is true even with noise and with a very simple reconstruction algorithm. As our main result we show that despite the algorithmic tractability of this problem, only very high degree polynomials are able to get any correlation with the root, in the same sense as Definition 4.

**Theorem 6** (Corollary 15 below). *Let $M$ be the transition matrix of a Markov chain on $[q]$ and suppose that $1 \leq k \leq q$ is such that $M^k$ is a rank-one matrix. For any function $f : [q]^L \to \mathbb{R}$ of Efron-Stein degree at most $2^{\lfloor \ell/(k-1) \rfloor}$ of the leaves $X_L$ and any prior $\nu$ on the root,*

$$\mathbb{E}[f(X_L) \cdot (\mathbb{1}(X_\rho = c) - \nu(c))] = 0.$$

*Remark* 7 (Tightness). For fixed $k$, this result is tight up to the base of the exponent. When $M$ satisfies the assumption of Theorem 5, there is a function on a constant-degree subtree to recover the root and (by Fourier expansion) this is a polynomial of degree $e^{O(\ell)}$.

To interpret this result, observe that $N = d^\ell$, so taking $d$ a constant, the Theorem shows that polynomials of degree even $N^c$ for an explicit constant $c = c(d, k) > 0$ fail to get any correlation with the root label. In comparison, in the previously mentioned contexts in the low-degree polynomials literature, the threshold for polynomials of degree $O(\log N)$ matches the conjectured threshold for polynomial time algorithms (see e.g. Hopkins and Steurer 2017; Hopkins 2018; Kunisky et al. 2019) and polynomials of degree $N^c$ correspond to conjectural thresholds for subexponential time algorithms (see e.g. Ding et al. 2019; Bandeira et al. 2020).

**A consequence: subexponential sample complexity lower bound for the RBF kernel.** As a consequence of our main result, we can analyze the behavior of kernel regression methods in our model. Kernel ridge regression is one of the canonical methods for solving supervised learning problems, including classification problems (see e.g. Muthukumar et al. 2021). In many high-dimensional settings, it is believed that the function learned using a standard kernel (e.g. Gaussian or polynomial) is essentially a low-degree polynomial. Standard results in learning theory (Shalev-Shwartz and Ben-David 2014) imply that kernel ridge regression with a Gaussian/RBF (Radial Basis Function) kernel in $N$ dimensions can learn a degree $\ell$ polynomial on the hypercube or sphere using roughly $O(N^\ell)$ samples. Establishing lower bounds on KRR is generally *much harder*. In certain particularly tractable settings (e.g. data from the uniform distribution on the hypercube) it has been recently shown explicitly that kernel regression (only) learns a low-degree polynomial (Ghorbani et al. 2021; Mei et al. 2021).

It seems plausible to guess that kernel ridge regression with a standard kernel will require subexponentially many samples of (leaf label, root label) pairs in order to learn to predict the root. We

---

[6]Note that in the single-sample case ($m = 1$), the information available to the algorithm would only be count statistics, which we know are insufficient for reconstruction below the KS threshold (Mossel 2004).

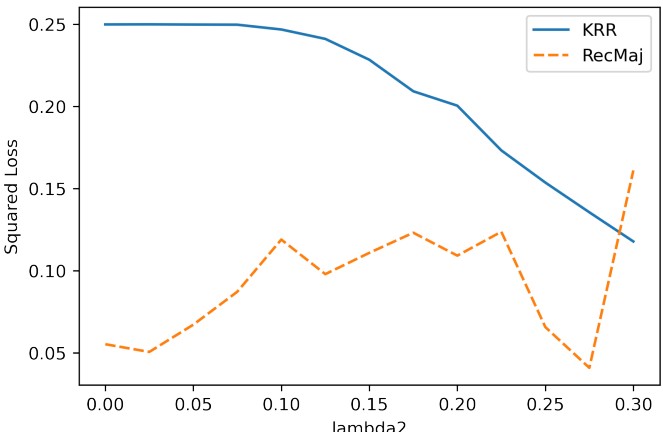

Figure 1: RBF Kernel Ridge Regression (KRR, blue line) test error in squared loss for predicting the root from leaves with data generated by broadcast operator $M_{\lambda_2} := (1 - \lambda_2)M_0 + \lambda_2 I$ and varying $\lambda_2$. $M_0$ is from (1), and it can be directly checked that $\lambda_2$ is the second eigenvalue of $M_{\lambda_2}$. The tree is 10-ary with depth 3, and the prior is $\nu = (0.5, 0.5, 0)$. RecMaj (orange dotted line) is a baseline estimator which generalizes the one used in proof of Theorem 5. Consistent with Theorem 8, the output of KRR fails to correlate with the root coloration when $\lambda_2 = 0$ (since 0.25 is the *null risk*, the squared loss for the optimal constant predictor), even though RecMaj correlates significantly with the root for all values of $\lambda_2$. In fact, KRR fails to correlate for all values of $\lambda_2$ up to around 0.1, suggesting that the failure of KRR and low-degree polynomials should extend beyond $\lambda_2 = 0$.

are able to verify this prediction in the case of the popular RBF (Radial Basis Function) kernel. See Section C.1.1 for formal notation and background on kernel ridge regression.

**Theorem 8** (Theorem 20 below). *Let $M$ be the transition matrix of a Markov chain on $[q]$ and suppose that $1 \leq k \leq q$ is such that $M^k = \pi\pi^T$ is a rank-one matrix, and suppose that $\pi$ has at least two nonzero entries. Then the for any color $c \in [q]$ and prior $\nu$ for the root coloration $X_\rho$, the following is true. Given $m$ i.i.d. samples $(x_1, y_1), \ldots, (x_m, y_m)$ from the broadcast model on the $d$-ary tree with $N$ leaves and broadcast channel $M$, where $x_i$ is a one-hot encoded vector of leaf colorations and $y_i = 1(X_\rho = c) - \nu(c)$ is the centered indicator of the leaf coloration, we have that for any bandwidth $\sigma \geq 0$ and ridge parameter $\lambda \geq 0$, for $w$ the output of ridge regression in RKHS space with those parameters and feature map $\varphi$, that with probability at least $1 - \delta$*

$$\frac{\mathbb{E}_{x_0, y_0}[y_0\langle w, \varphi(x_0)\rangle]}{\sqrt{\mathbb{E}_{x_0, y_0}[y_0^2]}} = O(\sqrt{1/N})$$

*provided that $m/\delta = O(e^{N^a})$ where $a = a(M, d) > 0$ is independent of the depth of the tree.*

This establishes a new and illustrative example where KRR performs poorly in high dimensions, even though the ground truth is a relatively "simple" and the labels are closely related to the structure of the input data. Note that the conclusion implies that $\mathbb{E}[(y_0 - \langle w, \varphi(x_0)\rangle)^2] \geq (1 - O(1/\sqrt{N}))\mathbb{E}[y_0^2]$, i..e. kernel ridge regression does not significantly outperform the constant zero estimator ("null risk") unless it is given at least a subexponential number of samples. Also, as with Remark 7 this result is tight up to the power of the exponent $c$, since a subexponential degree polynomial exists which predicts the root well and it can provably be learned with subexponential number of samples by KRR (Shalev-Shwartz and Ben-David 2014).

In Figure 1, we test kernel ridge regression in a simulation in both the case $\lambda_2 = 0$ and $\lambda_2 > 0$: consistent with our result, KRR fails to beat the null risk when $\lambda_2 = 0$; interestingly, it also fails for moderately small values of $\lambda_2$ as well, which is related to the Open Problem we discuss later. In the figure, KRR is performed using 2000 i.i.d. samples of $(x, y)$ pairs with $x$ the one-hot encoded leaf colorations and $y$ the centered indicator that the root color is 1, as in Theorem 8. Bandwidth and ridge penalty are selected via grid search on a validation set. The results for the baseline (RecMaj) are averaged over 16000 samples.

**Results for reconstruction with an unknown tree.** Formally, we consider the following variant of the generative model which is a variant of models in the phylogenetics literature. Briefly, in this model we generate $m$ i.i.d. realizations of the broadcasting model, where the tree is random and the prior on the root is biased towards a random root label. We include a parameter $\epsilon \geq 0$ which can be used to add noise to the final output the of model, just as above.

**Definition 9** ($\epsilon$-Noisy Repeated Broadcast Model on Random Tree). Let $\ell \geq 1, d \geq 2, m \geq 1, q \geq 1, \epsilon \geq 0$ and let $M$ be a Markov chain on $[q]$. Define $R = R_{\ell,d,m,M,\epsilon}$ by the following process:

1. Sample $Y^* \sim Uni([q])$, and $\tau \sim Uni(S_N)$ is a random permutation. Let $T$ be the $d$-ary tree on the set of leaves ordered by $\tau$, so e.g. vertices $\tau(1)$ and $\tau(2)$ are siblings in $T$.

2. Sample $X^{(1)}, \ldots, X^{(m)}$ i.i.d. from the $\epsilon$-noisy broadcast process (see Definition 3) on $T$ with prior $(2/3)\delta_{Y^*} + (1/3)Uni([q])$ and transition matrix $M$, where $\delta_{Y^*}$ is a delta distribution on $Y^*$. Let $\mathbb{X} = (X_L^{(1)}, \ldots, X_L^{(m)})$.

The goal of the learning algorithm in the unknown tree model is this: given $m$ samples of the leaves of the broadcast process, encoded in $\mathbb{X}$, reconstruct the root label $Y^*$ which the prior is biased towards[7]. We discuss the reasons for defining the model this way: 1. The permutation $\tau$ ensures that the coordinates of $X_L^{(i)}$ behave in a symmetric way, or equivalently that the order of those coordinates is not semantically meaningful; observe that if we omitted it, then the first $d$ coordinates would always be neighbors in the tree. This is standard in the phylogenetics literature (Steel 2016) and this kind of symmetry is also assumed in the literature on low-degree polynomial hardness, see e.g. discussion in Holmgren and Wein 2020; in sparse PCA this is analogous to how the support of the planted sparse vector is chosen uniformly at random among size-$k$ subsets. 2. The choice that root assignments are drawn from a tilted/biased distribution is different from the previous literature motivated by phylogenetics, where the root value is generally sampled fresh each time. This does not have a significant effect on how the algorithms used to estimate the tree work. The reason for our setup is to allow for straightforward comparison between SQ and low-degree polynomial models. If the root value was sampled from an unbiased measure each time, it would not make sense for an SQ algorithm to estimate it, since SQ has no concept of individual samples.

To be formal, we define the Statistical Query VSTAT oracle analogue of $R$ in the usual way (Feldman et al. 2017). The oracle is defined conditional on $Y^*$ and the tree $T$, so the order of leaves in the tree will be consistent between different calls to the oracle. As a reminder, vector-valued queries are implemented in the SQ model by querying each coordinate of the vector individually.

**Definition 10** ($VSTAT(m)$ Oracle). Let $Y^*, \tau, T$, and $M$ be as in Definition 9. Conditional on $Y^* = y^*$ and the tree $T = t$, we define $VSTAT(m)$ to be an arbitrary oracle which given a query function $\varphi : [q]^L \to [0,1]$, returns $p + \zeta_\varphi$ where $p := \mathbb{E}_R[\varphi(X^{(1)}) \mid Y^* = y^*, T = t]$ where $\zeta_\varphi$ is arbitrary (can be adversarily chosen) such that $|\zeta_\varphi| \leq \max\left(\frac{1}{m}, \sqrt{\frac{p(1-p)}{m}}\right)$.

We now state our results in this model. Just as in the known tree case, there is a relatively simple algorithm which achieves nearly optimal performance in this setting when $\lambda_2(M) = 0$ and $d$ is a large constant, and furthermore this algorithm can straightforwardly be implemented in the SQ model described above. Establishing this requires proving a new result in tree reconstruction, since (for example) the setting $\lambda_2(M) = 0$ which we care about rules out the use of Steel's evolutionary distance (see e.g. Steel 2016; Moitra 2018) commonly used in reconstruction algorithms in phylogeny, as Steel's distance is only well-defined for nonsingular phylogenies, and some kinds of tree models with singular matrices are actually computationally hard to learn (Mossel and Roch 2005).

**Theorem 11** (Theorem 30 below). *Suppose $M$ is a the transition matrix of a Markov chain with pairwise distinct rows, i.e. for all $i, j \in [q]$ the rows $M_i$ and $M_j$ are distinct vectors, and suppose $\lambda_2(M) = 0$. There exists $d \geq 1$ and $\epsilon > 0$ so that the following result holds true for the complete $d$-ary tree with any depth $\ell \geq 1$. For any $\delta > 0$, there exist a polynomial time algorithm with sample complexity $m = poly_M(\log N, \log(1/\delta))$ from the $\epsilon$-noisy repeated broadcast model (Definition 9) which with probability at least $1 - \delta$: 1) outputs the true tree $T$ (equivalently, the true permutation $\tau$), 2) outputs $\hat{Y}$ such that $\hat{Y} = Y^*$. Also, this algorithm can be implemented using a $VSTAT(m)$ oracle with $m = poly_M(\log(N/\delta))$ and polynomially many queries.*

---

[7]We could also consider the model where the root label is always $Y^*$. The soft bias we consider is nicer for minor technical reasons, and seems natural given we allow to add noise elsewhere in the model.

The lower bound for polynomials also applies here, just like in Theorem 6. Since it is very similar, we leave the formal statement to the appendix (Theorem 27).

## 1.3 Further Discussion

**Related work: complexity of reconstruction on trees.** Our work follows a line of previous work which identified the Kesten-Stigum threshold as a potential complexity barrier in the context of the broadcast model on trees. The work (Mossel 2016) showed that algorithms that do not use correlation between different features (named "shallow algorithms") cannot recover phylognies above the Kesten-Stigum threshold, where other ("deeper") algorithms can do so efficiently; the motivation in (Mossel 2016) was to find simple data models where depth is needed for inference. More standard complexity measures were studied in Moitra et al. 2020 who obtained a number of results on the circuit complexity of inferring the root in the broadcast process. They conjectured that below the Kesten-Stigum threshold, inferring the root is $NC1$-complete and proved it for one specific chain satisfying $\lambda_2 = 0$. Although there are some connections between low-degree polynomials and certain circuit classes (Linial et al. 1993), the results of this work and (Moitra et al. 2020) are incomparable and the techniques for establishing the lower bound are very different. Finally, we note the work (Jain et al. 2019) which studied the power of message-passing algorithms on finite alphabets: they proved such algorithms fail to recover all the way down to the Kesten-Stigum threshold, even in the simplest case of the binary symmetric channel with $q = 2$.

**Message passing vs. low-degree polynomials.** One of the attractive properties of the class of low-degree polynomials is that it generally captures the power of a constant (or sufficiently slowly growing) number of iterations of message-passing algorithms such as BP, AMP, and Survey Propagation (see e.g. Bresler and Huang 2021 and Appendix A of Gamarnik et al. 2020), which is interesting since a constant number of steps of these algorithms are indeed useful for many statistical tasks. On the other hand, in our models, belief propagation (which computes the exact posterior) succeeds with high probability whereas low-degree polynomials fail. This is not a contradiction: in our setting, BP requires $\Theta(\log N)$ iterations for the messages to pass from the leaves to the root and this is (as our main result shows) too large to simulate with low-degree polynomials.

**SQ and Low-Degree Polynomials.** The recent work (Brennan et al. 2021) established sufficient conditions for predictions to match between the Statistical Query (SQ) and low-degree polynomial heuristic, in a general setting. Nevertheless, in the unknown tree setting we consider above we saw that SQ algorithms perform significantly better than low-degree polynomials. The results of (Brennan et al. 2021) cannot be immediately applied to our setting, because we have phrased the problem as an estimation problem instead of a testing (a.k.a. distinguishing) problem; however, this is itself not the reason for the discrepancy as we could rephrase our problem in terms of testing the color of the root. Instead, the reason seems to be due to the "niceness condition" needed for their theory to apply. They show that the niceness condition will be satisfied for noise-robust problems when the "null distribution" in the testing problem is a product measure. Our setup is indeed noise robust (see Theorem 5). However, if we rephrased our problem as a testing one the null distribution will be a graphical model (with no bias at the root) and not a product measure.

Recently there has also been interest in understanding lower bounds against kernel learning algorithms (including polynomial kernels), in part motivated by connections to neural networks, and this involves connections to the SQ framework. See e.g. (Kamath et al. 2020) and references within. These methods can, for example, prove strong lower bounds against learning parities with kernel ridge regression since parities have large SQ-dimension. See also (Koehler and Risteski 2018) for another example where polynomial degree lower bounds were established for a Bayesian inference task, though only polylogarithmic in the dimension.

**Noise robustness and learning parities.** It was shown in (Janson and Mossel 2004) that the KS threshold is sharp for *robust reconstruction*, where we recall from above that (by definition) robust reconstruction on the $d$-ary tree with channel $M$ is possible if *for all* noise levels $\epsilon \in (0, 1)$, $\epsilon$-noisy reconstruction is possible. At first glance, this appears similar to the idea in the low-degree polynomials literature that the model should be *slightly noisy* to rule out the example of learning parities (which can be solved in the noiseless setting by Gaussian elimination, but not when there is noise). In fact the two notions are quite different: the robust reconstruction result shows that reconstruction

becomes impossible for *large noise levels* $\epsilon > \epsilon_*$ above a critical threshold $\epsilon_*$, whereas for small (but fixed) $\epsilon > 0$ the $\epsilon$-noisy reconstruction problem often *remains solvable* — see Theorem 5. Our examples are fundamentally different to the parity example: (1) for the unknown tree version of our model, we showed that the problem is solvable with an SQ oracle whereas parities are well-known to be hard for SQ (Blum et al. 2003), (2) relatedly, the algorithms which solve our problems are not "algebraic" in nature, and (3) our results hold irrespective of adding a small amount of noise, whereas learning parities with any constant amount of noise is conjecturally hard (Valiant 2012). Altogether, we can think of these results as suggesting a new, more nuanced picture of low degree vs. robustness to noise. Whereas before the main dichotomy in the computational complexity of inference literature has been between zero noise and any noise, in our model reconstruction algorithms such as BP can tolerate a small amount of noise, but fail when the noise level crosses some critical threshold. At least in our setting with $|\lambda_2(M)| = 0$, low-degree polynomials appear to capture the latter "large noise" behavior instead of the "small noise" difficulty of the problem.

**Open Problem.** What happens when $\lambda_2(M) \neq 0$? It is natural to wonder if the Kesten-Stigum threshold $d|\lambda_2|^2 = 1$ is sharp for low-degree polynomial reconstruction, analogous to how it is sharp for robust reconstruction. Our main lower bound result (Theorem 6) is consistent with this intuition. Also consistent with this intuition, our simulation result Figure 1 suggests that Kernel Ridge Regression may continue to fail for small but nonzero values of $\lambda_2(M)$. See Appendix B for a more formal statement and more discussion of this important question.

## 2 Technical Overview

The detailed proofs of all results are given in the Appendix. Here, we explain the high-level proof ideas, which we believe are relatively clean and conceptual.Before proceeding, we give the following concrete example of a Markov chain $M_0$ with $\lambda_2(M_0) = 0$ and $q = 3$:

$$M_0 = \begin{bmatrix} 0.5 & 0 & 0.5 \\ 0.25 & 0.5 & 0.25 \\ 0 & 1 & 0 \end{bmatrix}, \qquad M_0^2 = \begin{bmatrix} 0.25 & 0.5 & 0.25 \\ 0.25 & 0.5 & 0.25 \\ 0.25 & 0.5 & 0.25 \end{bmatrix}. \tag{1}$$

Since $M_0^2$ is rank one, it must be the case that $\lambda_2(M_0) = 0$.

**Failure of low-degree polynomials (Theorem 6).** We want to show that any low-degree polynomial $f$ of the leaves of the broadcast tree fails to correlate with the root. In general, it may be very difficult to compute the maximal correlation among all low-degree polynomials; what makes it possible in our case is that the correlation is exactly zero. If $c \in [q]$ is a color and $\nu$ is the prior at the root, we want to show $\mathbb{E}[f(X_L)(1(X_\rho = c) - \nu(c))] = 0$. (Recall $L$ is the set of leaves and $X_L$ the leaf colorations.) The first step is to use linear of expectation to break $f(X)$ into monomials: more formally, if $f(X) = \sum_{|S| \leq D} f_S(X)$ is the Efron-Stein decomposition for a polynomial of degree $D$, then to show the goal it clearly suffices to show

$$\mathbb{E}[f_S(X_L)(1(X_\rho = c) - \nu(c))] = 0.$$

Crucially, the monomial $f_S$ is a function which depends only on a set of at most $D$ leaf colorations $X_D$. Therefore, the result follows if we can show those leaves by themselves are independent of the root coloration. This is shown by performing an *iterative trimming* procedure on the minimal subtree spanned by the root and the leaves in $S$: every time there is an isolated path of length $k$ (where $M^k$ is rank one: $k = 2$ in the example above) all information is lost from the start of the path to its end. Using this idea and some elementary combinatorics, we can prove that if $|S|$ is small, the trimming procedure will delete everything, and so the root is indeed independent of these leaves.

**Failure of RBF Kernel Ridge Regression (Theorem 8.)** This result builds on the low-degree polynomials result. First, we show that if the bandwidth parameter in the kernel is taken too small, then the output of Kernel Ridge Regression (KRR) is close to zero on a new test point and so it fails to learn anything. Otherwise, we can directly show that any function with a substantial high degree polynomial component has large RKHS norm. We can also construct an interpolator of the training data which has much smaller RKHS norm, by showing that every training sample has a small "fingerprint" which uniquely identifies it and is detectable with a low-degree polynomial. It then follows that whatever the output of KRR is, it must have a small RKHS norm and cannot correlate with the true regression function.

**Success of noise robust reconstruction using "high degree" algorithms (Mossel and Peres 2003).** We briefly explain why noise robust reconstruction is possible with simple and computationally efficient algorithms. The key is to consider the case of a depth 1 tree: because the rows of $M$ are distinct, if the degree of the tree is a sufficiently large constant, then by the Law of Large Numbers the empirical distribution of its children will be close to the row of $M$ corresponding to the state of the parent, letting us reconstruct the parent with say $99.9\%$ probability of success. Given this, it is not too hard to argue this argument works recursively and in the presence of a small adversarial noise. Note that this algorithm recursively integrates *global* information on the tree across multiple scales — in contrast, the lower bound used the fact that low-degree polynomials can only aggregate information between small sets of variables in a limited (linear) way.

**Unknown tree results (Theorem 11 and Theorem 27).** The lower bound for low-degree polynomials in this setting can be reduced to the previous low-degree polynomial lower bound, which leaves proving that efficient (and SQ) algorithms can successfully solve this problem. Once we reconstruct the tree, we can run any algorithm for reconstructing the root given the leaves, e.g. the one described just above or BP. As far as reconstructing the tree, we first explain how to reconstruct the first layer. We prove that the joint distribution of any two leaves has enough information to tell us if they are immediate neighbors, which determines the location of all of their parents in the tree (such a test is easy to construct if we look at the *generalized eigenvectors* of $M$). Now that the bottom layer of the tree structure is determined, we use the fact that we have very good estimates of their parents colorations using the algorithm described before. Crucially, since that algorithm's accuracy guarantee for reconstructing the internal node's colors is very strong and does not decay as we go further and further up the tree, we can indeed apply this argument recursively to get the whole tree.

Implementing this algorithmic approach in the SQ framework is straightforward: at the end of the day it is based on computing the joint distributions of pairs of (estimated) vertex colorations, and those are all averages over the data. On the other hand, note that this method very strongly relies on the ability of an SQ algorithm to make *adaptive queries*, since the queries made are based on the partially reconstructed tree structure, which is unknown to the algorithm before it starts.

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
