# A  Organization of the Appendix

In the Appendices we give full proofs of all results; there is no dependence on the Technical Overview as all the information there will be repeated here in more detail. In Appendix B we state more formally the open problem described in the intro. In Appendix C we prove the results for known trees: in particular, this includes the main lower bound result, which is the failure of low degree polynomials for recovering the root; we also show how to deduce the RBF kernel lower bound using this. This is also where the RecMaj algorithm from Figure 1 is formally explained. In Appendix D we prove the results in the setting with an unknown tree; the main technical step is showing how to reconstruct the tree when $\lambda_2 = 0$ using a sample-efficient algorithm, which can be straightforwardly implemented in SQ.

# B  Open Problem: General Broadcast Chains

*Question* 12 (Kesten-Stigum is sharp for Low-Degree Polynomials?). Suppose that $d$ and transition matrix $M$ are such that $d|\lambda_2(M)|^2 < 1$, i.e. we are below the Kesten-Stigum threshold. Let $D_N$ be an arbitrary function of $N = d^\ell$ such that $D_N = O(\log N)$ as $\ell \to \infty$. Is it true that the degree-$D_N$ maximum correlation between the broadcast process at the leaves $X_L$ and the root $X_\rho$ in the sense of Definition 4 is asymptotically zero, i.e. $\liminf_{\ell \to \infty} \text{Corr}_{\leq D_N} = 0$? Equivalently, is it true that

$$\lim_{\ell \to \infty} \inf_{c \in [q]} \max_{deg(f) \leq D_N, \mathbb{E}[f(X_L)^2]=1} \sup \mathbb{E}_{\mu_\ell}[f(X_L)(1(X_\rho = c) - \nu(c))] = 0?$$

Here we make the common choice of looking at $\log N$ degree polynomials (see e.g. Hopkins and Steurer 2017; Kunisky et al. 2019), but any degree is interesting.

With the same intuition, we ask if a similar result to Theorem 8, the lower bound for kernel ridge regression, holds below the Kestum-Stigum threshold — see Figure 1 for related simulation results, which support the failure of KRR for small values of $\lambda_2$. We note that in our experiment the threshold where KRR starts to work is much closer to $d\lambda_2 = 1$. It is quite possible that this is a finite-depth effect since the experiment was done with a relatively shallow tree. Of course, if the sharp threshold is not the Kestum-Stigum threshold it would be extremely interesting to understand what the correct threshold is as a function of the broadcast model parameters.

# C  Known Tree: Upper and Lower Bounds

In this section, we prove a lower bound for arbitrary markov chains $M$ satisfying $\lambda_2(M) = 0$. From basic linear algebra (the existence of the Jordan Normal Form (Artin 2011)), we know that $\lambda_2(M) = 0$ if and only if $M^k$ is a rank one matrix for some $1 \leq k \leq q$, i.e. the Markov chain mixes perfectly in a finite number of steps. For concreteness, we give an example of such a chain with $k = 2, q = 3$ below.

*Example* 13 (Proof of Proposition 5, Mossel 2001). The following Markov chain on $q = 3$ states is a simple example of a chain with $\lambda_2(M) = 0$: we have

$$M = \begin{bmatrix} 0.5 & 0 & 0.5 \\ 0.25 & 0.5 & 0.25 \\ 0 & 1 & 0 \end{bmatrix}, \qquad M^2 = \begin{bmatrix} 0.25 & 0.5 & 0.25 \\ 0.25 & 0.5 & 0.25 \\ 0.25 & 0.5 & 0.25 \end{bmatrix}.$$

## C.1  Failure of Low-Degree Polynomials

**Theorem 14.** *Let $M$ be the transition matrix of a Markov chain on $[q]$ and suppose that $1 \leq k \leq q$ is such that $M^k$ is a rank-one matrix. Let $S$ be any subset of the leaves of the depth-$\ell$ complete $d$-ary tree $T = (V, E, \rho)$ with root $\rho$ and let $(X_v)_{v \in V}$ denote the broadcast process on $T$ with channel $M$. Let $S$ be an arbitrary subset of the leaf nodes of this tree. If $|S| < 2^{\lfloor \ell/(k-1) \rfloor}$, then $I(X_\rho; X_S) = 0$, i.e. $X_S$ is independent of the root value $X_\rho$.*

*Proof.* Assume for contradiction that $|S| < 2^{\lfloor \ell/(k-1) \rfloor}$ and $I(X_\rho; X_S) > 0$. Let $T_S$ be the minimal spanning subtree of $T$ containing the root node $\rho$ and all of the elements of $S$. (Equivalently, $T_S$ is the union of all of the root-to-leaf paths to $S$.)

Recall that in our convention, the edges of the tree $T$ are directed from the parent to the child. We say that $T_S$ contains an *isolated* length $k$ directed path if there exists adjacent nodes $u_0, \ldots, u_k$ contained in $T$ with $(u_i, u_{i+1}) \in E$ for all $0 \le i < k$, and such that nodes $u_1, \ldots, u_{k-1}$ all have degree 2 in $T_S$.

We show that we can reduce to the case where $T_S$ contains no isolated length $k$ directed paths. Otherwise, let $u_0, \ldots, u_k$ be as defined above and let $S_{u_k}$ be the subset of $S$ consisting of descendants of $u_k$ (note that by the definition of $T_S$, $S_{u_k}$ is nonempty). Observe that

$$I(X_\rho; X_S) \le I(X_\rho; X_S, X_{u_k}) = I(X_\rho; X_{S \setminus S_{u_k}}, X_{u_k})$$

where the last equality follows by the Markov property (all nodes in $S_{u_k}$ are descendants of $u_k$, so $X_{S_{u_k}}$ is independent of the root value $X_\rho$ conditionally on $X_{S \setminus S_u}, X_{u_k}$).

Next, by the chain rule for mutual information

$$I(X_\rho; X_{S \setminus S_{u_k}}, X_{u_k}) = I(X_\rho; X_{S \setminus S_{u_k}}) + I(X_\rho; X_{u_k} \mid X_{S \setminus S_{u_k}}) = I((X_\rho; X_{S \setminus S_{u_k}})$$

where the last equality follows from the fact that

$$I(X_\rho; X_{u_k} \mid X_{S \setminus S_{u_k}}) \le I(X_{u_0}, X_\rho; X_{u_k} \mid X_{S \setminus S_{u_k}}) = I(X_{u_0}; X_{u_k} \mid X_{S \setminus S_{u_k}}) = 0$$

where in turn the first equality follows from the Markov property ($X_\rho$ is independent of $X_{u_k}$ conditional on $X_{u_0}$ and $X_{S \setminus S_{u_k}}$) and the second equality follows because by the Markov property,

$$X_{S \setminus S_{u_k}} \to X_{u_0} \to X_{u_k}$$

is a Markov chain where the rightmost channel has transition matrix $M^k$, a rank-one matrix, so the conditional law of $X_{u_k}$ is the stationary measure of $M$ regardless of the value of $X_{u_0}$, hence $X_{u_k}$ is conditionally independent of $X_{u_0}$. Combining the above claims shows that

$$I(X_\rho; X_S) \le I(X_\rho; X_{S \setminus S_{u_k}})$$

where $|S \setminus S_{u_k}| < |S|$; by monotonicity of mutual information we in fact have

$$I(X_\rho; X_S) = I(X_\rho; X_{S \setminus S_{u_k}}).$$

Repeating this argument recursively reduces to the case where $T_S$ has no isolated length $k$ paths.

Finally, if $T_S$ has no isolated length $k$ paths then every internal node of $T_S$ is either: (a) at depth at most $k - 1$, or (b) has an ancestor at graph distance at most $k - 1$ away with degree at least 3. By induction, this implies that the number of nodes at depth $\ell'$ in $T_S$ is at least twice as large as the number of nodes at depth $\ell' - (k - 1)$. Since $S$ is the set of nodes in $T_S$ at depth $\ell$, this implies that

$$|S| \ge 2^{\lfloor \ell/(k-1) \rfloor}$$

which completes our proof by contradiction. $\qquad\square$

**Corollary 15.** *In the setting of the previous Theorem, for any function $f : [q]^L \to \mathbb{R}$ of Efron-Stein degree at most $2^{\lfloor \ell/(k-1) \rfloor}$ of the leaves $X_L$ and any prior $\nu$ on the root,*

$$\mathbb{E}[f(X_L) \cdot (\mathbb{1}(X_\rho = c) - \nu(c))] = 0.$$

*Proof.* By linearity of expectation and the Efron-Stein decomposition,

$$\mathbb{E}[f(X_L)X_\rho] = \sum_{S \subset L, |S| \le D} \mathbb{E}[f_S(X_L) \cdot (\mathbb{1}(X_\rho = c) - \nu(c))]\rangle] = 0$$

where the last equality used the previous Theorem and the fact $\mathbb{E}[\mathbb{1}(X_\rho = c)] = \nu(c)$. $\qquad\square$

### C.1.1 A consequence: failure of RBF kernel regression with oracle tuning

**Setting and notation.** We consider the performance of RBF kernel ridge regression (with arbitrary/oracle hyperparameter selection) for predicting the color of the root given the color of the leaves. As is customary, we encode the leaf vectors using a one-hot encoding, so the input to the regression is a list of i.i.d. samples $(x_i, y_i)_{i=1}^m$ where $x_i$ is the vector of one-hot encoded leaves, i.e. $(x_i)_{\ell,c} = \mathbb{1}(X_\ell = c)$, and for an arbitrary fixed color $c$, $y_i := \mathbb{1}(X_\rho = c) - \nu(c)$ is the centered indicator that the root is colored $c$.

**Background on Kernel Ridge Regression.** We remind the reader of some standard facts about kernel ridge regression and the Gaussian/RBF kernel — see (Shalev-Shwartz and Ben-David 2014) for a reference. Given a kernel $K(x, x')$, training points $x_1, \ldots, x_m$, and responses $y = (y_1, \ldots, y_m)$, the kernel ridge regressor with ridge parameter $\lambda$ is given by solving a linear equation

$$v = (\mathsf{K} + \lambda I)^{-1} y$$

where $\mathsf{K}_{ij} = K(x_i, x_j)$ is the kernel matrix, and the predicted response for a fresh data point $x_0$ is given by

$$\hat{y}_0 := \sum_{i=1}^{n} v_i K(x_i, x_0).$$

As is well-known, kernel ridge regression with ridge parameter $\lambda$ is equivalent to solving the ridge regression problem

$$\arg\min_{w} \sum_{i=1}^{n} (y_i - \langle w, \varphi(x_i) \rangle)^2 + \lambda \|w\|_2^2 \tag{2}$$

with feature vectors $\varphi(x)$ lying in a certain Hilbert space. Note that with this parameterization the prediction for fresh data point $x_0$ would just be $\hat{y}_0 = \langle w, \varphi(x_0) \rangle$ since $w$ is in the Hilbert space. In the case of the RBF kernel $K(x, y) = e^{-\|x-y\|_2^2/2\sigma^2}$, the corresponding feature map for $x \in \mathbb{R}^d$ is

$$\varphi(x) = e^{-\|x\|^2/2\sigma^2} \left( \frac{1}{\sigma^{2(n_1 + \cdots + n_d)}} \frac{x_1^{n_1} \cdots x_d^{n_d}}{\sqrt{n_1! \cdots n_d!}} \right)_{n_1, \ldots, n_d \geq 0} \tag{3}$$

so that $K(x, y) = \langle \varphi(x), \varphi(y) \rangle$. Note that $\|\varphi(x)\| = 1$ since $K(x, x) = e^0 = 1$.

**Proof of the lower bound.** We now proceed to prove the subexponential RBF sample complexity lower bound in our setting. For $\psi$ an element of the RKHS, define the orthogonal projection operator onto the space of degree $J$ and higher polynomials $P_{\geq J}$ by

$$(P_{\geq J} \psi)_{n_1, \ldots, n_d} := \begin{cases} 0 & \text{if } n_1 + \cdots + n_d < J \\ \psi_{n_1, \ldots, n_d} & \text{otherwise} \end{cases}.$$

From the definition, we first show that for large degree $J$ and bandwidth $\sigma$ not too tiny, $P_{\geq J}$ is very contractive when operating on feature embeddings $\varphi(x)$.

**Lemma 16.** *For any $x \in \mathbb{R}^d$ and $\varphi(x)$ as defined in (3) with bandwidth parameter $\sigma > 0$,*

$$\|P_{\geq J} \varphi(x)\|^2 \leq \frac{1}{\sqrt{J}} \left( \frac{e\|x\|^2}{J\sigma^2} \right)^J$$

*Proof.* First observe that

$$\frac{(\|x\|^2/\sigma^2)^j}{j!} = \sum_{n_1 + \cdots + n_d = j} \frac{1}{n_1! \cdots n_d! (\sigma^2)^{2j}} x_1^{2n_1} \cdots x_d^{2n_d}$$

by applying the multinomial theorem. Therefore,

$$\|P_{\geq J} \varphi(x)\|^2 = e^{-\|x\|^2/\sigma^2} \sum_{j=J}^{\infty} \frac{\|x\|^{2j}}{\sigma^{2j} j!} = e^{-\|x\|^2/\sigma^2} \frac{\|x\|^{2J}}{\sigma^{2J} J!} \sum_{j=0}^{\infty} \frac{\|x\|^{2j} J!}{\sigma^{2j} (J+j)!} \leq \frac{\|x\|^{2J}}{\sigma^{2J} J!}$$

and then the stated result follows from a nonasymptotic version of Stirling's approximation. $\square$

Next, we prove that there exists a relatively low-degree and low-RKHS norm polynomial which perfectly interpolates the training data, by showing that with high probability every sample has a small and unique "fingerprint" given by looking at a small set of well-separated leaves.

**Lemma 17.** *Let $M$ be the transition matrix of a markov chain on $[q]$ and suppose that $1 \leq k \leq q$ is such that $M^k = \pi\pi^T$ is a rank-one matrix, and suppose that $\pi$ has at least two nonzero entries. Then if $S$ is a set of leaves of distance at least $2k$ from each other and $X_1, \ldots, X_m$ are i.i.d. random vectors generated by the broadcast process with transition matrix $M$, the probability that there exists $i, j \in [m]$ such that $(X_i)_S = (X_j)_S$ is at most $\binom{m}{2} \delta^{|S|}$ where $\delta = \delta(M) \in (0, 1)$ is a constant depending only on $M$.*

*Proof.* First, let $X, X'$ be independent samples of the leaves from the generative model and let $c = c(M) > 0$ be such that the stationary distribution $\pi$ has at least two entries of size at least $c$. For $S$ a set of leaves of distance at least $2k$ from each other, we have by the Markov property that the entries of $X_S$ are independent from each other conditional on the values of the markov process $X_v$ for all vertices $v$ at height $k$ above the leaves; we see then that the conditional law of the leaves $X_S$ is $\pi^{\otimes S}$ which does not depend on $X_v$, so in fact the leaves $X_S$ are unconditionally distributed according to the product measure $\pi^{\otimes S}$. Then by independence,

$$\Pr(X_S = X'_S) = \prod_{i \in S} \Pr(X_i = X'_i) \leq (1 - c)^{|S|}$$

where in the last step we used that regardless of the value of $X_i$, $X'_i$ has a probability at least $c$ of being different from it. $\square$

**Lemma 18.** *For $x \in \{0, 1\}^d$ with $\sum_i x_i = p$, and $S \subseteq [d]$ and $b_S \in \{0, 1\}^d$ arbitrary, there exists $w = w(p, S)$ of (Hilbert space) norm*

$$\|w\|^2 \leq 2^{|S|} e^{p/2\sigma^2} \max\left\{1, \sigma^{2|S|}\right\} \sqrt{|S|!}$$

*such that*

$$\langle w, \varphi(x) \rangle = 1(x_S = b_S).$$

*Proof.* Observe that

$$1(x_S = b_S) = \prod_{i \in S} [b_i x_i + (1 - b_i)(1 - x_i)]$$

which for fixed $b$, expands into a sum of at most $2^{|S|}$ many monomials of degree at most $|S|$ and with coefficient 1. Representing this expanded polynomial in the RKHS, using (3), then leads to the stated norm bound. $\square$

We show that the overlap between two independent samples of the leaves from the model concentrates exponentially with a subgaussian tail:

**Lemma 19.** *Let $M$ be the transition matrix of a markov chain on $[q]$ and suppose that $1 \leq k \leq q$ is such that $M^k = \pi\pi^T$ is a rank-one matrix. Then if $X_L, X'_L$ are two independent random vectors of leaf colorations generated by the broadcast process on the $d$-ary tree with $N = |L|$ leaves and $x_L, x'_L$ are the corresponding one-hot encodings, we have that*

$$\Pr\left(\left|\frac{1}{N}\langle x_L, x'_L\rangle - \|\pi\|_2^2\right| > t\right) \leq 2e^{-cNt^2}$$

*where $c = c(M, d) > 0$ is a constant not depending on $N$.*

*Proof.* First, observe that if $N$ is smaller than $d^k$, this bound can be proved trivially by shrinking $c$, so henceforth we assume $N$ is larger than this. By the law of total probability, it is sufficient to prove the desired bound conditional on the colors $X_V, X'_V$ where $V$ is the set of vertices at height $k$ above the leaves, and similar to the proof of Lemma 17 we observe by the Markov property that this makes the color of the set of children of any particular $v \in V$ independent of the colors of all non-children of $v$. This means that $\langle x_L, x'_L \rangle$ a sum of bounded independent random variables, and because $M^k = \pi\pi^T$ we have that its expectation is $\|\pi\|_2^2 N$, so the result follows immediately from Hoeffding's inequality (Vershynin 2018). $\square$

**Theorem 20.** *Let $M$ be the transition matrix of a markov chain on $[q]$ and suppose that $1 \leq k \leq q$ is such that $M^k = \pi\pi^T$ is a rank-one matrix, and suppose that $\pi$ has at least two nonzero entries. Suppose that $m/\delta \leq e^{cN^\epsilon}$. Then given $m$ i.i.d. samples $(x_1, y_1), \ldots, (x_m, y_m)$ from the broadcast model on the $d$-ary tree with $N$ leaves and broadcast channel $M$, we have that for any bandwidth $\sigma \geq 0$ and ridge parameter $\lambda \geq 0$, for $w$ the output of ridge regression in RKHS space with those parameters, that with probability at least $1 - \delta$*

$$\frac{\mathbb{E}_{x_0, y_0}[y_0 \langle w, \varphi(x_0) \rangle]}{\sqrt{\mathbb{E}_{x_0, y_0}[y_0^2]}} = O(\sqrt{1/N})$$

*provided that $m/\delta = O(e^{N^\epsilon})$ where $\epsilon = \epsilon(M, d) > 0$ is independent of $N$ (equivalently, independent of the depth of the tree).*

*Proof.* As usual, we will use that $N$ can be assumed larger than a fixed absolute constant without loss of generality. The proof is via case analysis on the bandwidth parameter $\sigma$.

First we make an argument which covers the case of small bandwidth parameter $\sigma$. Note that for any $i$, $\|x_i\|^2 = N$ almost surely since there are $N$ leaves and each leaf is one-hot encoded. By Lemma 19 and the union bound, with probability at least $1 - \delta/4$ for any $i \neq j$ in $[m]$ we have

$$\|x_i - x_j\|_2^2 = 2N - 2\langle x_i, x_j \rangle \geq 2(1 - \|\pi\|_2^2)N - O_{M,d}(\sqrt{N\log(m/\delta)})$$

so

$$\mathsf{K}_{ij} = e^{-\|x_i - x_j\|_2^2/2\sigma^2} \leq \exp\left([-(1 - \|\pi\|_2^2)N + O_{M,d}(\sqrt{N\log(m/\delta)})]/\sigma^2\right).$$

It follows that there exists $c_2 = c_2(M, d) > 0$ such that if $\sigma \leq c_2 N^{1/2-\epsilon/4}$, then $(\mathsf{K})_{ij} \leq e^{-N^{\epsilon/3}}$ for $i \neq j$ and so by Gershgorin's disk theorem and the fact that the diagonal of $\mathsf{K}$ is all-ones, $\|\mathsf{K} - I\|_{OP} \leq 1/N$. Hence for the Kernel Ridge solution $v = (\mathsf{K} + \lambda I)^{-1}y$ we have $\|v\| \leq 2\|y\| \leq 2\sqrt{m}$.

Consider a fresh test set of independently sampled pairs of leaf and root colorations $(x'_1, y'_1), \ldots, (x'_{ms}, y'_{ms})$ where $s := N\log(2/\delta)$. Observe by Hoeffding's inequality that with probability at least $1 - \delta/4$,

$$\left|\frac{1}{ms}\sum_{i=1}^{ms}\left(\sum_{j=1}^{m}v_j K(x_j, x_0)\right)^2 - \mathbb{E}_{x_0}\left[\left(\sum_{j=1}^{m}v_j K(x_j, x_0)\right)^2\right]\right| = O(\sqrt{\log(2/\delta)/s})$$

where $x_0$ is a fresh one-hot encoded vector of leaf colorations sampled from the same distribution and where we used the fact that $\|v\| \leq 2\sqrt{m}$ and $K(\cdot, \cdot) \leq 1$ to show that over the randomness of $x_0$, $\left|\sum_{j=1}^{m}v_j K(x_j, x_0)\right| \leq 2\sqrt{m}$ almost surely, which we used in order to apply Hoeffding's inequality. By repeating the argument used to show the off-diagonal entries of $\mathsf{K}$ are small, we have with probability at least $1 - \delta/4$

$$\frac{1}{ms}\sum_{i=1}^{ms}\left(\sum_{j=1}^{m}v_i K(x_j, x_i)\right)^2 \leq me^{-N^{\epsilon/2}},$$

hence by the triangle inequality we have with probability at least $1 - \delta$ that

$$\mathbb{E}_{x_0}\left[\left(\sum_{i=1}^{m}v_i K(x_i, x_0)\right)^2\right] \leq me^{-N^{\epsilon/2}} + O(\sqrt{\log(2/\delta)/s})$$

and recalling $s = N\log(2/\delta)$ gives the result in this case.

Now we cover the remaining set of bandwidth parameters where $\sigma > c_2 N^{1/2-\epsilon/4}$. By the combination of Lemma 17 applied with $|S| = C_M \log(m/\delta)$ and Lemma 18, we have that there exists $w$ such that for every $x_i$

$$\langle w, \varphi(x_i) \rangle = y_i$$

and

$$\|w\| \leq (m/\delta)^{C'_M} e^{N/4\sigma^2} \sigma^{C_M \log m/\delta} \sqrt{(C_M \log m/\delta)!}. \tag{4}$$

It follows that the output of KRR with any ridge parameter $\lambda \geq 0$ has norm at most the rhs of (4) (otherwise, replacing the output with $w$ would shrink the norm without decreasing the training error in (2)). Next, by Lemma 16 we have that for any $x$ and degree $J$

$$\langle P_{\geq J}w, \varphi(x) \rangle = \langle w, P_{\geq J}\varphi(x) \rangle$$
$$\leq \|w\|\|P_{\geq J}\varphi(x)\|$$
$$\leq \|w\|\frac{1}{\sqrt{J}}\left(\frac{e\|x\|^2}{J\sigma^2}\right)^J = \|w\|\frac{1}{\sqrt{J}}\left(\frac{eN}{J\sigma^2}\right)^J$$

so taking as in Corollary 15 $J = 2^{\lfloor \ell/(k-1) \rfloor} = N^\epsilon$ where this equation defines $\epsilon$ and using that

$$N/J\sigma^2 = N^{1-\epsilon}/\sigma^2 = O(\sigma^{-\epsilon/(1-\epsilon/2)}),$$

we have that for any $w$ satisfying (4),

$$|\langle P_{\geq J}w, \varphi(x)\rangle| \leq \|w\|(c_3/\sigma^{\epsilon/(1-\epsilon/2)})^{N^\epsilon}$$
$$\leq (m/\delta)^{C'_M}e^{N^{\epsilon/2}/4c_2}\sigma^{C_M \log m/\delta}\sqrt{(C_M \log m/\delta)!}(c_3/\sigma^{\epsilon/(1-\epsilon/2)})^{N^\epsilon} = O((1/\sigma)^{N^{\epsilon/2}}).$$

Since by Corollary 15 and Cauchy-Schwarz we have that

$$\mathbb{E}_{x_0,y_0}[y_0\langle w, \varphi(x_0)\rangle] = \mathbb{E}_{x_0,y_0}[y_0\langle P_{\geq J}w, \varphi(x_0)\rangle] \leq \sqrt{\mathbb{E}_{x_0,y_0}[y_0^2]}\sqrt{\mathbb{E}_{x_0,y_0}[\langle P_{\geq J}w, \varphi(x_0)\rangle^2]}$$

combining this with the bound on $|\langle P_{\geq J}w, \varphi(x)\rangle|$ completes the proof. $\qquad\square$

## C.2 Success of noise-robust reconstruction using non-low-degree algorithms

Above we saw that when $|\lambda_2(M)| = 0$, very high degree polynomials are needed to get any estimate correlated with the root. Nevertheless, for "most" matrices $M$ with $|\lambda_2(M)| = 0$ and for degree $d$ sufficiently large as a function $M$ there exists a simple recursive and noise-robust method which witnesses the fact that reconstructing the root is possible. If one likes, this recursive function can trivially be expressed as a polynomial: then it will be a very high-degree polynomial that is nonetheless robust to noise.

The reason for the qualifier "most" in the discussion above is that there are some degenerate $M$ for which the task is clearly impossible: e.g. if $M$ is rank one (so it does not depend on its input). There are other similar examples, e.g. the chain on 3 states which deterministically transitions from state 1 to state 2, and such that at states 2 and 3 the chain flips a fair coin to transition to either state 2 or 3. With this clarified, we can now state the known positive result for reconstruction.

**Theorem 21** (Theorem 6.1 of Mossel 2004)**.** *Suppose $M$ is a the transition matrix of a Markov chain with pairwise distinct rows, i.e. for all $i, j \in [q]$ the rows $M_i$ and $M_j$ are distinct vectors. Then there exists $d_0 = d_0(M)$ such that for all $d \geq d_0$, reconstruction is possible on the $d$-ary tree.*

A variant of the condition in this Theorem gives a tight characterization of Markov chains where reconstruction is possible on the infinite $d$-ary tree for sufficiently large $d$, see Theorem 2.1 of Mossel and Peres 2003.

By revisiting the proof of Theorem, we get the following slightly more precise result which we will use in later sections. This result shows that for any desired accuracy $\delta$, for sufficiently large degrees $d$ there exists a noise-tolerant estimator $f$ which reconstructs the root correctly with probability at least $1 - \delta$ uniformly of the color of the root.

**Theorem 22** (Proof of Theorem 2.1 of Mossel and Peres 2003)**.** *Suppose $M$ is a the transition matrix of a Markov chain with pairwise distinct rows, i.e. for all $i, j \in [q]$ the rows $M_i$ and $M_j$ are distinct vectors. Let $\delta \in (0, 1)$ be arbitrary. There exists $d_0 = d_0(M, \delta), \epsilon > 0$ such that for all $d \geq d_0$, $\epsilon$-noisy reconstruction is possible on the $d$-ary tree and furthermore there exists a polynomial-time computable function $f = f_{M,\ell}$ valued in $[q]$ such that*

$$\max_{c\in[q]}\Pr(f(X'_L) \neq X_\rho \mid X_\rho = c) < \delta$$

*where $X'_L$ is the $\epsilon$-noisy version of $X_L$ (see Definition 3).*

*Proof sketch.* As explained above, this result follows from examination of the proof of Theorem 2.1 in Mossel and Peres 2003. For the reader's convenience, we summarize the main idea of the proof.

In the base case of a depth 1 tree, reconstruction of the root with probability at least $1 - \delta$ is possible provided $d$ is a suitably large constant, because by basic large deviations theory (Sanov's Theorem (Dembo and Zeitouni 2010)) the empirical distribution of the children will concentrate around the row of $M$ corresponding to the root label (which by assumption is distinct from all of the other rows). This procedure is also robust to a small amount of noise, which handles the case where $\epsilon > 0$ and in fact even if the $\epsilon$ proportion of children assigned labels by the noise process choose their labels adversarially. When doing the induction, the result of the reconstruction process at lower levels of the tree can therefore (by conditional independence) be modeled as the true values with a small amount of adversarial noise and this allows the same argument to show that at each level each vertex is recovered correctly with probability at least $1 - \epsilon$ (where we take $\epsilon := \delta$). $\qquad\square$

*Remark* 23 (RecMaj in Figure 1). The RecMaj algorithm in Figure 1 corresponds to the algorithm described in the above proof sketch: i.e. a recursive algorithm which to reconstruct the coloration of a vertex, looks at the reconstructions of its children, takes the empirical distribution, and picks the corresponding row of $M$ which is closest in $\ell_2$ norm.

### C.3 Low-Degree Polynomials succeed above the KS threshold

The Kesten-Stigum threshold is the sharp threshold for *count reconstruction* defined earlier. The definition of count reconstruction informally says that there is a nontrivial amount of mutual information between count statistics at the leaves and the value of the Markov Random Field at the root. To relate count reconstruction to low-degree polynomials, we use the following more precise result:

**Lemma 24** (Proof of Theorem 1.4 of Mossel and Peres 2003). *Suppose that $d|\lambda_2(M)|^2 > 1$. There exist coefficients $s_c \in \mathbb{C}$ for $c \in [q]$ such that the random variable*

$$S = \sum_{c \in [q]} s_c \#\{X_\ell = c : \ell \in L\}$$

*satisfies*

$$\mathbb{E}[S \mid X_\rho = c] = v_c$$

*where $v$ is an unit-norm eigenvector of $M$ in its second-largest eigenspace, i.e. achieving $\|Mv\| = |\lambda_2(M)|$, and such that*

$$\mathbb{E}[|S|^2 \mid X_\rho = c] \in [A, B]$$

*where $0 < A \le B$ are constants depending only on $d$ and $M$ (in particular, they are independent of the depth of the tree).*

As a consequence of this, we immediately obtain that low-degree polynomials (in fact, degree 1 polynomials) have nontrivial correlation with the root above the KS threshold, in the same sense as Definition 4.

#### C.3.1 A Question: Bayes-Optimal Reconstruction

We saw above that degree-1 polynomials of the leaves are sufficient to achieve nontrivial correlation with the root, provided that the model we consider is above the KS threshold. A natural question is whether higher degree polynomials have a significant advantage over degree-1 polynomials for estimating the value of the root. Relevant to this question, we recall the following result and conjecture from Mossel et al. 2014 which concerns noise-robust recovery with the Binary Symmetric Channel (equivalently, the Ising model on trees without external field):

**Theorem 25** (Theorem 3.2 of Mossel et al. 2014). *There exists an absolute constant $C \ge 1$ such that the following result is true. For $\theta \ge 0$ let*

$$M = \begin{bmatrix} (1+\theta)/2 & (1-\theta)/2 \\ (1-\theta)/2 & (1+\theta)/2 \end{bmatrix}$$

*and observe that $\lambda_2(M) = \theta$. If $d\theta^2 > C$, then for all $\epsilon < 1$ and $X'_L$ defined by the $\epsilon$-noisy broadcast model,*

$$\lim_{\ell \to \infty} d_{TV}(\mathcal{L}_{\mu_\ell}(X'_L = \cdot \mid X_\rho = 1), \mathcal{L}_{\mu_\ell}(X'_L = \cdot \mid X_\rho = 0))$$
$$= \lim_{\ell \to \infty} d_{TV}(\mathcal{L}_{\mu_\ell}(X_L = \cdot \mid X_\rho = 1), \mathcal{L}_{\mu_\ell}(X_L = \cdot \mid X_\rho = 0))$$

*in other words, if $\epsilon < 1$ is fixed then in the limit of infinite depth the probability of reconstructing the root correctly is the same as in the noiseless case $\epsilon = 0$.*

(Recall that the equivalence of the statement in terms of TV and in terms of maximum probability of reconstructing the root follows from the Neyman-Pearson Lemma Neyman and Pearson 1933.) This statement is conjectured to hold with $C = 1$ Mossel et al. 2014 and as explained there, is closely related to Bayes-optimal recovery in the stochastic block model. Based on this, we ask the following question:

*Question* 26. Do polynomials of degree $O(\log N)$ achieve asymptotically Bayes-optimal recovery with the above channel when $d\theta^2 > 1$? More precisely, does there exist a polynomial threshold function $f$ of degree $O(\log N)$ which asymptotically achieves

$$\Pr(f(X_L) = X_\rho) = (1 + o(1))\Pr(\mathrm{sgn}(\mathbb{E}[X_\rho \mid X_L] - 1/2) = X_\rho)$$

where the rhs is the error of the Bayes-optimal estimator.

It seems likely the answer to this question is positive. The reason for this is the following: (1) if the conjectured strengthening of Theorem 25 is true, then it implies that the combination of a majority vote up to some depth and $\omega(1)$ number of rounds of belief propagation achieves Bayes-optimal recovery, and (2) a constant or very slowly growing number of rounds of belief propagation can be simulated with low-degree polynomials (see Appendix of Gamarnik et al. 2020), and the threshold used in the majority vote should also be approximable by polynomials. We state the conjecture with $O(\log N)$ degree polynomials since this is informally considered to correspond to "polynomial time algorithms" in the low-degree framework (Hopkins 2018; Kunisky et al. 2019), but based on the above discussion it seems likely that a smaller degree than $O(\log N)$ is sufficient, e.g. any degree going to infinity with $N$ may be sufficient.

# D  Unknown Tree Setting

In this section, we show that for any channel $M$ satisfying the conditions of Theorem 22, i.e. such that for sufficiently large $d$ reconstructing the root is possible (in the known tree setting/in the usual sense), then in the unknown tree setting that a relatively simple algorithm succeeds at reconstructing the root with a polynomial number of samples, and this algorithm can be straightforwardly implemented in the SQ (Statistical Query) model with polynomial number of queries and error tolerance.

The key step in the algorithm for reconstructing the root is a method of reconstructing the tree, which lets us reduce to the known tree setting. This kind of problem has previously been extensively studied in the context of phylogenetic reconstruction with particular channels $M$ coming from biology, and for example algorithms with polynomial runtime and sample complexity are known in the case that $M$ is a nonsingular matrix (Mossel and Roch 2005). In the present context, we are very interested in the case of singular matrices (e.g. those with $\lambda_2(M) = 0$) so we cannot rely on existing results.

**Model.**   We remind the reader that in the unknown tree setting, we are in the model of Definition 9. This means that an unknown $Y^*$ is sampled from $Uni([q])$, and the algorithm seeks to reconstruct $Y^*$ given access to $m$ i.i.d. samples $X_L^{(1)}, \ldots, X_L^{(m)}$ of the leaves generated by the broadcasting process with root prior $(2/3)\delta_{Y^*} + (1/3)Uni([q])$, i.e. the root is biased/tilted towards the unknown $Y^*$. When we say the tree is "unknown" in this model, it means that the algorithm is not given a priori knowledge of the true order of the leaves, e.g. the algorithm does not know at the beginning whether coordinates 1 and 2 of $X_L^{(1)}$ correspond to siblings or to leaves far apart in the tree (this is completely analogous to the situation in phylogenetic reconstruction Steel 2016). In the definition of this model, this is modeled by shuffling the order of the leaves by an unknown permutation $\tau$; note that this order is kept consistent between each sample.

## D.1  Failure of low-degree polynomials

**Theorem 27.** *Let $M$ be the transition matrix of a Markov chain on $[q]$ and suppose that $1 \le k \le q$ is such that $M^k$ is a rank-one matrix. If $c \in [q]$ is arbitrary and $f$ is a polynomial with Efron-Stein degree strictly less than $2^{\lfloor \ell/(k-1) \rfloor}$, then*

$$\mathbb{E}_R[f(\mathbb{X})(\mathbb{1}(Y^* = c) - 1/q)] = 0$$

*where $R$ is as defined in Definition 9.*

*Proof.* Let $\nu(c) = 1/q$ for $c \in [q]$ denote the prior on $Y^*$.

By linearity of expectation and the definition of Efron-Stein degree, it suffices to show the result for functions $f$ of the form $f_{S_1}(X_L^{(1)}) \cdots f_{S_m}(X_L^{(m)})$ where $\sum_i |S_i| < 2^{\lfloor \ell/(k-1) \rfloor}$, where each

$f_{S_i}(X_L^{(i)})$ is a function only of the coordinates of its input in $S_i$. Since the samples $X^{(1)}, \ldots, X^{(m)}$ are conditionally independent given the value of $Y^*$, we have

$$\mathbb{E}_R\left[\left(\prod_{i=1}^m f_{S_i}(X_L^{(i)})\right)(\mathbb{1}(Y^* = c) - \nu(c))\right]$$

$$= \mathbb{E}_R\left[\mathbb{E}\left[\left(\prod_{i=1}^m f_{S_i}(X_L^{(i)})\right)(\mathbb{1}(Y^* = c) - \nu(c)) \mid Y^*\right]\right]$$

$$= \mathbb{E}_R\left[\left(\prod_{i=1}^m \mathbb{E}[f_{S_i}(X_L^{(i)}) \mid Y^*]\right)(\mathbb{1}(Y^* = c) - \nu(c))\right]$$

$$= \mathbb{E}_R\left[\left(\prod_{i=1}^m \mathbb{E}[f_{S_i}(X_L^{(i)})]\right)(\mathbb{1}(Y^* = c) - \nu(c))\right] = 0$$

where in the first equality we used the law of total expectation, in the second equality we used the aforementioned conditional independence, in the third equality we crucially used that by Theorem 14 the low-degree polynomial $f_{S_i}(X_L^{(i)})$ is independent of the root value and thus $Y^*$, and in the last step we used that $Y^* \sim \nu$ by definition. $\qquad\square$

### D.2 Reconstruction Algorithm

For $c \in [q]$, let $e(c)$ or $e_c$ denote the $q$th standard basis vector in $\mathbb{R}^q$. In both cases, the vector is a column vector.

**Lemma 28.** *Suppose that $\nu$ is a probability measure on $[q]$ and $\nu(c) > 0$ for all $c \in [q]$, then there exists a constant $\alpha = \alpha(M, \nu) > 0$ such that the following is true. Let $(X_u)_u \sim \mu$ for $u \in V$ be defined by the broadcasting process on $T = (V, E, \rho)$ with prior $\nu$ at the root and channels corresponding to $M : q \times q$ the transition matrix of an ergodic Markov chain. Then $\mu(X_u = c) > \alpha$ for all $u \in V$.*

*Proof.* Under the assumptions, there exists some $\beta > 0$ such that $\nu = \beta \pi_M + (1 - \beta)\nu'$ for $\nu'$ a probability measure. Because $\pi_M$ is the stationary distribution and the marginal law at any vertex $u$ is $\nu M^k$ for some $k \geq 0$, it follows that $\mu(X_u = c) > \beta \pi_M(c) \geq \min_c \beta \pi_M(c) =: \alpha > 0$. $\qquad\square$

**Lemma 29.** *Suppose that $u, v$ are two descendants of node $w$ at graph distance $k$ from $w$ and random variables $X_u, X_v, X_w$ follow the Markov process on trees $\mu$ with transition matrix $M : q \times q$. Then*

$$\mathbb{E}[e(X_u)e(X_v)^T] = (M^k)^T \Pi_w M^k$$

*where $\Pi_w : q \times q$ is a diagonal matrix with entries the marginal law of $X_w$, i.e. $(\Pi_w)_{cc} = \mu(X_w = c)$ for $c \in [q]$.*

*Proof.* Using the law of total expectation and using by the Markov property that $X_u$ and $X_v$ are conditionally independent given $X_w$, we have

$$\mathbb{E}[e(X_u)e(X_v)^T] = \mathbb{E}[\mathbb{E}[e(X_u) \mid X_w]\mathbb{E}[e(X_v)^T \mid X_w]] = \mathbb{E}[(e_{X_w}^T M^k)^T(e_{X_w}^T M^k)] = (M^k)^T \Pi_w M^k$$

where in the last equality we used the definition of $\Pi_w$ and the definition of the broadcast process in terms of the transition matrix $M$. $\qquad\square$

Based on this, we can recursively reconstruct the tree when the degree is sufficiently large. We note that for other channels like the BSC channel, tree reconstruction methods often handle internal nodes $u$ by computing majorities of the nodes under them, which gives an unbiased estimate of the spin $X_u$, but this technique is not applicable in our setting (it's unclear that unbiased estimators exist). Nevertheless, we show that applying the estimator from Theorem 22 can be used in a similar way, provided the degree $d$ is sufficiently large.

**Theorem 30.** *Suppose $M$ is a the transition matrix of a Markov chain with pairwise distinct rows, i.e. for all $i, j \in [q]$ the rows $M_i$ and $M_j$ are distinct vectors. If $|\lambda_2(M)| > 0$, additionally suppose that the prior on the root of the tree is the stationary distribution of $M$. There exists*

$d \geq 1$ *and* $\epsilon > 0$ *so that the following result holds true for the complete* $d$-*ary tree with any depth* $\ell \geq 1$. *For any* $\delta > 0$, *there exist a polynomial time algorithm with sample complexity* $m = poly_M(\log N, \log(1/\delta))$ *from the* $\epsilon$-*noisy repeated broadcast model (Definition 9) which with probability at least* $1 - \delta$:

1. *outputs the true tree* $T$ *(equivalently, the true permutation* $\tau$*)*

2. *outputs* $\hat{Y}$ *such that* $\hat{Y} = Y^*$.

*Also, this algorithm can be implemented in the Statistical Query (SQ) model using a* $VSTAT(m)$ *oracle with* $m = poly_M(\log(N/\delta))$ *and polynomial number of queries.*

*Proof.* Given that the algorithm can correctly output the true tree $T$, the fact that it outputs the correct root label follows straightforwardly from Theorem 22 by using the algorithm specified in that result to estimate the root in each sample, and then taking the majority vote over those samples (which will succeed with high probability provided we take $\Omega(\log(2/\delta))$ samples due to Hoeffding's inequality), and this can approach can also clearly be implemented in the SQ model (the SQ query is the robust reconstruction function of the leaves which outputs a vector, so we take the expectation of this and look at the largest entry of this vector). In the remainder of the proof, we show how to correctly output the true tree $T$ with high probability.

We first prove the result in the case that $\lambda_2(M) \neq 0$ and afterwards describe how to modify the argument straightforwardly when $\lambda_2(M) = 0$. Let $\varphi$ be a right eigenvector such that $M\varphi = \lambda_2\varphi$. We start by describing the algorithm which computes the estimated tree $\hat{T}$ from the bottom up: let $\alpha = \alpha(M, \delta) > 0$ be a parameter to be set later. Let $\hat{\mathbb{E}}[\cdot]$ denote the expectation over the empirical distribution of $m$ samples, so for any function $f$ we have $\hat{E}[f(X)] = \frac{1}{m}\sum_i f(X^{(i)})$.

1. Base case: for all leaves $u \neq v$ define $g(u,v) := |\langle \varphi, \hat{E}[e(X_u)e(X_v)^T]\varphi\rangle|$. Let $g_{max} = \max_{u \neq v} g(u,v)$ and set $u, v$ to be neighbors in $\hat{T}$ iff $g(u,v) \geq g_{max} - \alpha$. This constructs the first layer of the tree $\hat{T}$.

2. Recursive case: suppose that we have reconstructed the first $s \geq 1$ layers of the tree (from the bottom), and the current layer of the tree has more than one element. For each pair of internal nodes $u, v$ at the current level of the tree, let $S_u, S_v$ be the set of leaves under these nodes and let $g_s(u,v) := |\langle \varphi, \hat{E}[e(f_{M,\ell-s}(X_{S_u}))e(f_{M,\ell-s}(X_{S_v}))^T]\varphi\rangle|$ where $f_{M,\ell-s}$ is as defined in Theorem 22. Let $g_{max} = \max_{u \neq v} g(u,v)$ and set $u, v$ to be neighbors in $\hat{T}$ iff $g(u,v) \geq g_{max} - \alpha$. This constructs the next layer of the tree $\hat{T}$.

We now need to show that with total probability at least $1 - \delta$, $\hat{T} = T$. First we consider the behavior of the base case; for simplicity, we first describe the argument when $\epsilon = 0$. Observe that if $u$ and $v$ are siblings in $T$ at depth $\ell$ then by Lemma 29

$$\langle \varphi, \mathbb{E}[e(X_u)e(X_v)^T]\varphi\rangle = |\lambda_2|^2\langle \varphi, \Pi_{\ell-1}\varphi\rangle$$

where $\Pi_{\ell-1}$ is a diagonal matrix encoding the marginal law of $X_w$ for any $w$ at depth $\ell - 1$, and similarly, if $u$ and $v$ are not siblings then they are at graph distance at least 4 in $T$ so

$$\langle \varphi, \mathbb{E}[e(X_u)e(X_v)^T]\varphi\rangle \leq |\lambda_2|^4\langle \varphi, \Pi_{\ell-1}\varphi\rangle$$

which is smaller by a factor of $|\lambda_2|^2$. (Note, here we are using the fact that in the case $|\lambda_2| > 0$, we additionally assumed the prior at the root is stationary and so the marginal law at every depth in the tree is the stationary distribution.) Observe that by Hoeffding's inequality and the union bound we have that with probability at least $1 - \delta/n$ that in the base case step, every entry of the matrix $\hat{\mathbb{E}}[e(X_u)e(X_v)^T]$ for every pair of leaves $u \neq v$ is within additive error $O(\sqrt{\log(n/\delta)/m})$ of its expectation. It follows from this and Lemma 28 that if $\alpha = (1/C_M)(|\lambda_2|^2 - |\lambda_2|^4)$ for $C_M$ a sufficiently large constant depending only on $M$, $\epsilon$ is sufficiently small with respect to $\alpha$, and $m = \Omega_M(\log(n/\delta))$ then in the base case the algorithm computes neighbors correctly. Observe that at each layer, if the algorithm has correctly reconstructed $T$ in all previous layers then the sets $S_u$ for all nodes $u$ in this layer are deterministic functions of $T$, and hence so are the queries the

algorithm makes to $\hat{E}$. By a similar application of the union bound and Hoeffding's inequality as well as Theorem 22 and the assumption that $d$ is sufficiently large with respect to $M$ it follows that the algorithm succeeds at all subsequent layers as well.

Note that provided we take $\epsilon > 0$ is sufficiently small, we can show the base case of the argument will still succeed by using the triangle inequality, and the inductive step in the argument will succeed because of Theorem 22.

Finally, in the case that $\lambda_2(M) = 0$, we let $\varphi$ be a generalized eigenvector such that $M\varphi \neq 0$ but $M^2\varphi = 0$. Note that such a vector must exist because, 0 is an eigenvalue of algebraic multiplicity $q - 1$ as $M$ is ergodic and $\lambda_2 = 0$, and because our assumption on $M$ rules out the case that $M$ is rank one, so it's Jordan normal form must have at least one Jordan block with size at least 2 and this corresponds to the existence of such a generalized eigenvector $\varphi$. Now observe for such a $\varphi$ that if $u, v$ are siblings in $T$ at depth $\ell$ then

$$\langle \varphi, \mathbb{E}[e(X_u)e(X_v)^T]\varphi \rangle = \langle M\varphi, \Pi_{\ell-1} M\varphi \rangle$$

which by Lemma 28 is lower bounded by a constant $C'_M > 0$, while if $u, v$ are not siblings,

$$\langle \varphi, \mathbb{E}[e(X_u)e(X_v)^T]\varphi \rangle = 0.$$

Setting $\alpha = C'_M/2$ and defining the remaining constants similarly to above ensures the algorithm succeeds, by the same argument.

Note that in both the case $\lambda_2(M) \neq 0$ and $\lambda_2(M) = 0$, the algorithm is implemented by taking the expectation of certain functions over the samples, so it is straightforwardly implementable with SQ queries by replacing the empirical expectation with the VSTAT oracle. $\qquad\square$