# OpenReview forum: "Reconstruction on Trees and Low-Degree Polynomials"
_NeurIPS.cc/2022/Conference — NeurIPS 2022 Accept_

### Official Review · Reviewer_zt31 · 2022-07-10

**Rating:** 7
**Confidence:** 4
**Soundness:** 3 good
**Presentation:** 3 good
**Contribution:** 3 good

**Summary:**

This paper studies the problem of tree reconstruction on $d$-ary trees; the root of the tree is given a spin $X_\rho \sim \nu$, which is then propagated down to the leaves according to a Markov channel $M$. The problem is then, given the spins $X_L$ at the leaves, to recover the original root spin $X_\rho$. Several variants of this model are considered: with/without noise at the leaves, with the underlying tree known or not, or with several realizations of the same tree process.

The focus in this paper is on low-degree polynomial reconstruction: for which values of $M$ can $X_\rho$ be estimated by a low-degree polynomial in the leaves, i.e. a function of the form
$$ f(X_L) = \sum_{S\subset L, |S| = D} f_S(X_S) , $$
where $X_S$ is the subset of leaves in $S$. It is already known that when $d |\lambda_2(M)|^2 > 1$, a linear ($D = 1$) estimator suffices; on the other hand, general reconstruction using belief propagation is possible for almost all $M$.

The authors show that if $\lambda_2(M) = 0$, then no polynomial algorithm of degree $\leq N^c$, where $N$ is the number of leaves, can recover the true root spin. The proof is based on the property that $M^k$ is of rank 1 for some k, and hence the correlation between a vertex $x$ and its $k$-th ancestor is 0. As a corollary, they show that a kernel ridge regression method needs at least $e^{N^c}$ samples to learn the tree reconstruction problem, since it needs to approximate a polynomial of degree at least $N^c$.

The article also contains a positive result: for the case where the underlying tree is unknown (equivalently, where the leaves are known up to permutation), they show that for a fixed root spin $X_\rho$ and a polynomial number of samples from the tree process started at $X_\rho$, there exists a reconstruction algorithm that recovers $X_\rho$ better than random chance.

**Questions:**

- In Fig.1, is it a one-off run or are the results averaged over multiple samples ? It seems weird that the RecMaj algorithm is so inconsistent depending on $\lambda_2(M)$.
- Can those results be extended to similar tree distributions, with the same growth rate ? For example, using the br(T) value of Evans et al. '00.

**Limitations:**

The limitations have been adequately addressed.

**Strengths And Weaknesses:**

The tree reconstruction problem is ubiquitous in many inference problems (e.g. community detection), for which computational-to-statistical gaps are still fairly unexplained. It's interesting to see a low degree polynomial approach to this problem, which bridges the gap between the census reconstruction problem and BP approaches. The paper is overall well-written and easy to read; the introduced notions are clearly defined (with the notable exception of the VSTAT oracle), and the results are nicely presented. It is especially interesting that the impossibility result extends to $O(N^c)$ degrees, although this might just be a consequence of $\lambda_2(M) = 0$.

The main weakness of this paper, in my opinion, is its specificity: all proofs hinge on the specific properties that occur when $M^k$ is of rank one, which implies very string independence properties between tree nodes. The tree structure is similarly rigid, with only the $d$-ary tree considered. However, this is a good first step which I hope will inspire more work on this topic.

---

> ### Author Response · Authors · 2022-08-01
> **Thanks and responses to questions**
>
> We thank the reviewer for their feedback and answer questions below:
>
> > In Fig.1, is it a one-off run or are the results averaged over multiple samples ? It seems weird that the RecMaj algorithm is so inconsistent depending on \lambda_2(M)
>
> Good question — the results for the RecMaj algorithm were averaged over 16000 samples so this is really the accuracy it achieves. (We will add this information to the figure caption in the next version.) Why the curve should look the way it does is not obvious, but we expected \lambda_2(M) to be strongly correlated with the performance of KRR and not so correlated with RecMaj so at least it is consistent with this intuition.
>
> > Can those results be extended to similar tree distributions, with the same growth rate ? For example, using the br(T) value of Evans et al. '00.
>
> It should be straightforward to extend the negative results to different tree topologies. For positive results (e.g. information-theoretic possibility of reconstruction), it should be possible but more complicated regularity conditions would be required than just looking at br(T). For example, if M^k is rank one and the top of the tree is a length k path, then the leaves will have no mutual information with the root even though br(T) can be large.

---

### Official Review · Reviewer_Rmuy · 2022-07-11

**Rating:** 5
**Confidence:** 1
**Soundness:** 3 good
**Presentation:** 1 poor
**Contribution:** 3 good

**Summary:**

This paper studies the problem of reconstruction on trees through low degree polynomials. The authors show that there exists simple tree models in which nontrivial reconstruction of the root value is possible in polynomial time, and if the tree is unknown but given samples with correlated root assignments, nontrivial reconstruction is possible with a statistical query algorithm. The paper also provide a result related to RBF kernel ridge regression for predicting root coloration. An open question about low degree polynomials and the KS threshold is also proposed.

**Questions:**

- Some acronyms are used without introduction, for example, MCMC and CSP.
- It would be great if the authors could provide some high level interpretation of their results for those who are not familiar with this line of work.

**Limitations:**

Not applicable.


**Strengths And Weaknesses:**

The topic of this paper is completely out of my area, and any technical comments I make will probably be unfair to the authors.

Regarding organization: I can hardly follow the paper. Partly it is because the topic is out of my area. However in the current shape, everything including introduction, preliminaries, definitions, theorems, remarks, are all mixed into the two massive sections. While I understand there might be a lot of contents in the paper, I think the presentation can definitely be improved.

I also don't know what are the exact contributions in this paper. For example Theorem 5 (Mossel and Peres 2003) appears under Section 1.2 Our Results. Is it a new result, or from a prior work?

---

> ### Author Response · Authors · 2022-08-01
> **Thanks and responses to questions**
>
> We thank the reviewer for their suggestions on the presentation and answer questions below:
>
> > I also don't know what are the exact contributions in this paper. For example Theorem 5 (Mossel and Peres 2003) appears under Section 1.2 Our Results. Is it a new result, or from a prior work?
>
> To clarify the discussion about Theorem 5, we did intend to credit this result to Mossel and Peres 2003 and will revise accordingly. The formal Theorem 22 and its proof are included for completeness, as the fact that this estimator is robust to additional noise naturally follows from their argument but is not part of the original theorem statement. The reason we included Theorem 5 here (with the MP ‘03 reference) was as a way to introduce the context for Theorem 6, our main result. The reviewer makes a good point that since all the other stated theorems in the section are our new contributions, it is potentially confusing when skimming — we will edit the text here to further reduce confusion.
>
> > It would be great if the authors could provide some high level interpretation of their results for those who are not familiar with this line of work.
>
> We hope the following brief summary helps clarify the context of this work and our contribution and will edit the paper accordingly.
>
> A large body of recent work attempts to predict “computational-to-statistical gaps”: situations where it is impossible for polynomial time algorithms to learn anything from the data, even though computationally inefficient (“information-theoretic”) algorithms can succeed at the same task. Such gaps appear in a large variety of problems (for example, sparse PCA) and one of the most popular and successful heuristics to predict these gaps is to look at low-degree polynomials, usually of degree O(log(N)) in problem size N, as an easier-to-analyze proxy for computationally efficient algorithms. (Standard tools like NP-hardness are not useful for analyzing average-case problems due to complexity-theoretic reasons, so such heuristics are very important.)
>
> The main contribution of this paper is to show that for a canonical and well-studied statistical task (reconstruction on trees) the natural low-degree polynomial heuristic does not predict the correct threshold — it predicts the problem is hard when it is in fact solvable by polynomial time algorithms. Formally our main result, Theorem 6, shows that only polynomials of N variables with degree at least N^{c} for c > 0 a constant can solve this statistical task. As consequences of the main result, we can show that the related method of RBF kernel regression also fails at this task unless provided exp(N^c) many samples (Theorem 8) and that for a variant of this task, another heuristic from the literature called “Statistical Query Algorithms” (SQ)  predicts the correct threshold while low-degree polynomials still fail (Theorem 11). The last result is significant because prior work has focused on equivalences between SQ and low-degree predictions (see paper “Statistical Query Algorithms and Low-Degree Tests Are Almost Equivalent”), and our work is illustrating a concrete example where the equivalence must break down.
>
> > Some acronyms are used without introduction, for example, MCMC and CSP
>
> Good point, we will edit the text to introduce these acronyms (Markov Chain Monte Carlo, Constraint Satisfaction Problem) for improved clarity.

---

### Official Review · Reviewer_APcw · 2022-07-20

**Rating:** 8
**Confidence:** 3
**Soundness:** 3 good
**Presentation:** 3 good
**Contribution:** 3 good

**Summary:**

Investigates the problem of tree reconstruction (from leaves to root) through low degree polynomials.


**Questions:**

None.

**Limitations:**

The main limitation I can see is that the tree structure that makes this analysis go through is somewhat limited.


**Strengths And Weaknesses:**

Low-degree polynomials as a computational model used to study (in)tractability of learning/inference problems seems to be a useful and important model, so studying when it does/doesn't reflect the wider class of polynomial-time algorithms is very important. The results in this paper provide an important insight in this direction. The assumptions are pretty carefully justified, and the authors are largely working in standard models.

---

### Official Review · Reviewer_bRPZ · 2022-07-21

**Rating:** 7
**Confidence:** 3
**Soundness:** 3 good
**Presentation:** 3 good
**Contribution:** 3 good

**Summary:**

This paper studies the effectiveness of using the restricted computational model class of low-degree polynomials for assessing statistical-computation gaps for high-dimensional statistical inference problems.  Using the problem of reconstruction on trees, the authors identify problem settings for which average-case reconstruction is impossible using low-degree polynomials yet reconstruction is possible with computationally efficient methods.

**Questions:**

N/A

**Limitations:**

yes

**Strengths And Weaknesses:**

Strengths (major)
- significance: Information-computation tradeoffs for high dimensional statistical inference problems are important.  There are several restricted computational models that have been used to study gaps for different problems (planted clique, sparse PCA, etc.).  Low-degree polynomials are a prominent model.  The authors show that there are problems where thresholds identified using low-degree polynomials do not match thresholds for existence of computationally efficient methods.
- The extension of the lower bound to investigate when kernel ridge regression would fail is interesting.
- The authors provide substantial discussions.

Strengths (minor)
- While work is theoretical, the authors include an experiment (Fig 1) investigate performance of low-degree polynomials for small but non-zero $\lambda_2$, suggesting that the inaccuracy of low-degree polynomials to capture computational hardness may not narrowly occur on the limit case (of $\lambda_2=0$) analytically studied in the paper.

Weaknesses
- I did not identify any major weaknesses.

Very minor notes:
- line 30, acronym CSP not defined
- line 78 – the acronym for statistical query ‘SQ’ is used multiple times before being defined
- Line 95 - Notation $c$ not introduced yet, why not $x_r$ for realization of root variable?
- Line 103 – $\rho$, a subscript appearing in the conditioning event, is not defined.
- Lines 117 and 122 missing parenthesis
- line 189 – a constant $c$ for $N^c$ is discussed, but that is different than the $c$ as the root variable realization, right? If so, I’d suggest not overloading notation.
- line 210 capitalize ‘markov’
- Line 212 should ‘Suppose that …’ be there?  Line 218 has a condition on $m/\delta$ and in 212 the notation $m$, $\delta$, $\epsilon$, and $c$ not yet specified
- Line 217-218, I don’t think the notation $\varphi(\cdot)$ was introduced yet

---

> ### Author Response · Authors · 2022-08-01
> **Thanks for the feedback**
>
> We thank the reviewer for their detailed feedback and will make the improvements suggested.
>
> > line 189 – a constant c for N^c is discussed, but that is different than the c as the root variable realization, right? If so, I’d suggest not overloading notation.
>
> Yes that’s right, these are unrelated and we will change the notation.
>
> > Line 212 should ‘Suppose that …’ be there?
>
> You are right, this is redundant with line 218 and should be removed.

---

### Meta-Review · Area_Chair_o8B6 · 2022-08-30

**Recommendation:** Accept
**Confidence:** Certain

**Metareview:**

This paper studies using low-degree polynomials for analyzing statistical/computational gaps for high-dimensional inference problems and identify average-case settings that exhibit this gap.  This is a nice paper and above the bar, though it perhaps appeal to only a theoretical audience.

**Award:**

No

---

### Decision · Program_Chairs · 2022-09-14

Accept